



# A global view on stratospheric ice clouds: assessment of processes related to their occurrence based on satellite observations

Ling Zou[1], Sabine Griessbach[1], Lars Hoffmann[1], and Reinhold Spang[2]

[1]Jülich Supercomputing Centre (JSC), Forschungszentrum Jülich, Jülich, Germany
[2]Institute of Energy and Climate Research (IEK-7), Forschungszentrum Jülich, Jülich, Germany

**Correspondence:** Ling Zou (l.zou@fz-juelich.de; cheryl_zou@whu.edu.cn)

**Abstract.**

Ice clouds play an important role in regulating water vapor and influencing the radiative budget in the atmosphere. In this study, stratospheric ice clouds (SICs) and stratospheric aerosols from the Cloud-Aerosol Lidar and Infrared Pathfinder Satellite Observations (CALIPSO), deep convection and gravity waves from Atmospheric Infrared Sounder (AIRS) observations and
tropopause temperature from ERA5 are analyzed to investigate their long-term variation and processes potentially related to the formation of SICs on the global scale.

SICs with cloud top heights 0.25 km above the first tropopause are mainly detected over the tropical continents. SICs associated with the double tropopause events, where the cloud top is between the first and second thermal tropopause, are mostly located in midlatitudes (between $25° - 60°$). The seasonal cycle and the inter-annual variability of SIC frequencies from 2007 to
2019 show that high SIC frequencies are mainly observed south of the equator from November to March, and at $10° N - 20° N$ from July to September. At mid- and high latitudes, more SICs are observed from December to May in the northern hemisphere and in the southern hemisphere during May to October.

Relations between SICs and first tropopause temperature, deep convection, gravity waves, and stratospheric aerosol were analyzed, respectively, on a global scale. Positive correlations between SIC frequencies and deep convection, gravity waves, and
stratospheric aerosol and an inverse correlation between SIC frequency and tropopause temperature were observed worldwide. Overlaps of high correlations/anti-correlations were detected over tropical continents, i, e., tropical South America, equatorial Africa, and western Pacific, suggesting a combined effect of tropopause temperature, deep convection, gravity waves, and stratospheric aerosol on SIC occurrence in these regions. Over Central America, North America, the Asian Monsoon, and mid- and high latitudes deep convection and gravity waves present a strong correlation with the occurrence of SICs, individually or
interdependently.

Regional analyses demonstrated specific relations of tropopause temperature, deep convection, gravity waves, and stratospheric aerosol with SICs at a finer scale. Low tropopause temperature and high occurrence frequency of stratospheric aerosol show strong correlations with high frequencies of SICs over the Indo-Pacific Warm Pool, tropical South America, and equatorial Africa. Deep convection and gravity waves have the strongest correlation with occurrence frequency of SICs over the
Asian Monsoon and the North American Monsoon. Gravity waves and tropopause temperature are highly correlated with SIC occurrence over South America and the northern Atlantic. Moreover, the El Niño phenomenon in 2009-2010 and 2015-2016



coincides with low SIC occurrences over the Indo-Pacific Warm Pool. High stratospheric aerosol loads related to volcanic eruptions (Puyehue-Cordón Caulle and Nabro in 2011) and wildfires (over the United States and Canada in 2017) are closely related to high occurrence frequencies of SICs.

We investigated the global distribution and long-term variation of SICs and present a global view of relations between SIC occurrence and tropopause temperature, deep convection, gravity wave activity, and stratospheric aerosol. This work provides a better understanding of the physical processes and climate variability of SICs.

## 1    Introduction

Stratospheric ice clouds (SICs) play an important role in regulating the water vapor in the upper troposphere and lower strato-
sphere (UTLS), i. e., ice cloud formation and sedimentation may dehydrate the UTLS (Jensen and Pfister, 2004; Schoeberl and Dessler, 2011; Schoeberl et al., 2019), while injection of convective clouds and sublimation of ice in the lower stratosphere would hydrate stratosphere (Dinh et al., 2012; Jain et al., 2013; Avery et al., 2017). Thin ice clouds in the UTLS region produce radiative heating by trapping outgoing longwave radiation, while thick ice clouds cause radiative cooling in the atmosphere (Zhou et al., 2014; Lolli et al., 2018). SICs are also important indicators for better understanding the vertical temperature struc-
ture in the UTLS, transport between troposphere and stratosphere, intensity and dynamics of deep convection (Liou, 1986; Corti et al., 2006; Mace et al., 2006; Jensen et al., 2011; Kärcher, 2017). Therefore, understanding the microphysical and macrophysical properties of SICs is of importance for global atmosphere modeling and future climate prediction.

Global occurrence of high altitude ice clouds is about $20-40\%$ over the world (Liou, 1986; Wylie et al., 1994, 2005). The earliest discoveries of stratospheric ice clouds were reported in Murgatroyd and Goldsmith (1956) and Clodman (1957) from
in-situ observations. Since then, more and more studies have demonstrated the existence of SICs from in-situ measurements, satellite measurements and ground-based lidar observations (Wang et al., 1996; Keckhut et al., 2005; De Reus et al., 2009; Dessler, 2009; Spang et al., 2015; Bartolome Garcia et al., 2021). For example, 7 % of observations with the cloud top above the first tropopause were detected from lidar observations over the Site Instrumental de Recherche par Télédétection Atmosphérique (SIRTA) between 2002 and 2006 (Noël and Haeffelin, 2007). Six encounters in total 90 encounters with ice clouds
in the tropical lower stratosphere were observed from Forward Scattering Spectrometer Probe (FSSP-100) and Cloud Imaging Probe (CIP) measurements (De Reus et al., 2009). 5-day lasting SICs at 18.6 km on March 2014 were found over Gadanki from the ground-based Mie lidar observations and space-borne observations (Sandhya et al., 2015). Several cases of ice clouds were discovered above convective anvils reaching up to the lower stratosphere from the Geostationary Operational Environmental Satellite (GOES) and the Next Generation Weather Radar (NEXRAD) program Weather Surveillance Radar-1988 Doppler
(WSR-88D) network (Homeyer et al., 2017).

On a global scale, the worldwide distribution of SICs is detected from the Cloud-Aerosol Lidar and Infrared Pathfinder Satellite Observations (CALIPSO) measurements (Pan and Munchak, 2011; Zou et al., 2020). More SICs are observed over the tropics than mid- and high latitudes. The SICs are more often distributed over tropical continents with frequencies as high





as 24 % to 36 %. With increasing evidence for the existence and occurrence of SICs, potential driving forces for the formation

and maintenance of ice clouds in the UTLS attract more attention.

Atmospheric thermodynamics, dynamics, and aerosol properties are critical features for stimulating the formation of ice clouds and inducing a change of ice cloud (Haag and Kärcher, 2004). Ice cloud formation in the UTLS shows a significant dependence on cold ambient temperature (Holton and Gettelman, 2001), direct injection or outflow from deep convection (Jensen et al., 1996; Massie et al., 2002), large-scale upwelling of moist air parcels (Brewer, 1949; Pfister et al., 2001), and

temperature perturbations induced by atmospheric waves and wave breaking (Boehm and Verlinde, 2000; Podglajen et al., 2018).

Nucleation of ice crystals is in favor of cold temperatures in the UTLS. For example, 5-day lasting SICs at 18.6 km over Gadanki on March 2014 were attributed to a wave-induced cold temperature anomaly (Sandhya et al., 2015). A large scale cirrus cloud on 27–29 January 2009 over the Eastern Pacific observed by CALIPSO was caused by the lower temperature

induced by an extratropical intrusion (Taylor et al., 2011). The temperature controls the formation and variability of ice clouds (Tseng and Fu, 2017).

Deep convection leads to ice clouds formation directly from ice injection, anvil outflow, and indirectly from radiative and dynamic cooling associated with updrafts and waves (Homeyer et al., 2017). Tropopause-penetrating convection and large convection-related UTLS winds are more closely associated with the detection of SICs (Homeyer et al., 2017). 47 % cirrus

clouds were observed at 10-15 km on Manus Island, Papua New Guinea, in 1999 initiated from deep convection based on ground-based (millimeter cloud radar (MMCR) and Geosynchronous Meteorological Satellite (GMS)) data (Mace et al., 2006). A close association of cirrus clouds and deep convection activities in the TTL was found based on a 2-year dataset from CALIPSO and CloudSat satellites, as high deep convection frequency are detected together with high cirrus cloud frequency (Sassen et al., 2009). During the Deep Convective Clouds and Chemistry (DC3) experiment, stratospheric cirrus were observed

at altitudes of 1 – 2 km above the tropopause on May–June 2012 over the continental United States evolved from enhanced deep convection (Homeyer et al., 2014), which is facilitated by double tropopause events (Peevey et al., 2012, 2014).

Perturbations of the temperature fields induced by gravity waves and wave breaking have substantial impacts on the occurrence of ice clouds (Schoeberl et al., 2015; Jensen et al., 2016; Dinh et al., 2016; Wang et al., 2016). Atmospheric waves significantly modulate the occurrence and maintenance of ice clouds in the tropical tropopause layer (TTL) at an altitude range

of 14-18 km over the Pacific based on observations from the Global Hawk aircraft in the Airborne Tropical TRopopause EXperiment (ATTREX) (Kim et al., 2016). Wave-induced cooling of air parcels has a strong influence on the cirrus cloud occurrence. Similar results have been presented by Chang and L'Ecuyer (2020), with about 80 % of the cirrus clouds detected in the cold phase of gravity waves and wave-induced air parcel cooling process at altitude above 14.5 km over tropics during January 2007 to December 2013 from CALIPSO and 2C-ICE measurements. By using a cloud model - Wisconsin Dynamic and Microphys-

ical Model (WISCDYMM), Wang et al. (2016) found that internal gravity wave breaking is the source of stratospheric cirrus layers on 23 December 2009 in Argentina in CALIPSO measurement.

Uplifted aerosol particles, such as sulfate aerosol, organic aerosol, and dust from volcanic eruptions or biomass burning, are effective ice nuclei for cirrus cloud formation and variation (Lohmann et al., 2003; Lee and Penner, 2010; Jensen et al.,





2010; Froyd et al., 2010; Cziczo et al., 2013). Cirrus cloud formation and a five times enhanced ice crystal concentration was

demonstrated by microphysical simulations assuming volcanic aerosol at temperatures below about -50°C (Jensen and Toon, 1992). The stratospheric sulfur aerosol and cirrus reflectance show a strong inverse correlation in 2001-2011 from Moderate Resolution Imaging Spectroradiometer (MODIS) observations (Friberg et al., 2015). From MOSAiC (Multidisciplinary drifting Observatory for the Study of Arctic Climate), Ohneiser et al. (2021) observed 30 km high aerosol and cloud layers over high altitudes in the northern hemisphere from October 2019 to May 2020 and found out that those cloud layers were generated by

the major wildfire events in July and August 2019. Atmospheric aerosol show an impact on the occurrence and variability of ice clouds, while the influence of aerosol on their microphysical properties remains highly uncertain.

The large-scale atmospheric disturbances, e. g., El Niño–Southern Oscillation (ENSO), Madden-Julian Oscillation (MJO), and the stratospheric quasi-biennial oscillation (QBO) affect the occurrence of ice cloud in the UTLS (Collimore et al., 2003; Eleftheratos et al., 2007; Inai et al., 2012; Liess and Geller, 2012). In the warm phase of ENSO (the El Niño condition), a

cold tropopause and more ice clouds were found in the UTLS over the mid-Pacific, whereas a warm tropopause and less ice clouds were found over the western Pacific and Indonesia (Massie et al., 2000; Avery et al., 2017). The MJO also plays an important role in influencing the occurrence of ice clouds at altitudes of $\sim 15 - 18$ km (Inai et al., 2012). Virts and Wallace (2010) investigated the relation of ice clouds in TTL with MJO and ENSO. They found that MJO-related deep convection modulates ice cloud frequency in the TTL, i. e., higher frequencies of ice clouds are observed over equatorial continents when

convection over the Pacific is enhanced. Phases of ENSO regulate the zonal shift of peak frequency of ice clouds in the tropics that more ice clouds can be found over the central Pacific region in the warm ENSO phase (El Niño) and more detected over the Maritime Continent in the cold phase (La Niña). Eleftheratos et al. (2011) explored the contribution of ENSO and QBO on the interannual variability of cirrus clouds from 1985-2005 over the tropical Pacific Ocean from the International Satellite Cloud Climatology Project (ISCCP) data. They found that the largest impact on cirrus cloud variability over the eastern and

western Pacific is generated by ENSO. Moreover, the flow of moist air from the tropical upper troposphere to the extra-tropical stratosphere along isentropic levels is also important for the occurrence of SICs. The quasi-isentropic transport of high humidity air was found to be a source for the occurrence of SICs over northern middle and high latitudes (Spang et al., 2015) based on the Cryogenic Infrared Spectrometers and Telescopes for the Atmosphere (CRISTA) measurements in August 1997.

The occurrence and maintenance of ice clouds shows a significant dependence on atmospheric dynamics and thermal struc-

ture. The regional and global occurrence and distribution of SICs have been examined in previous works. However, studies on potential formation mechanisms of high-altitude ice clouds have typically been limited by short-term observations with specific factors and mainly over small regions. To better understand the ice clouds detected in the lower stratosphere, we used 13 years (2007-2019) cloud data from CALIPSO to revisit the global stratospheric ice clouds and explore their relations with tropopause temperature, deep convection, gravity waves, and stratospheric aerosol. In addition, regions of SIC hotspots were selected and

analyzed in combination with the above factors and ENSO and QBO indices to investigate potential mechanisms of occurrence of SICs. Information on data and detection methods for SICs, stratospheric aerosol, deep convection, and gravity waves are presented in Section 2. Global SIC occurrences, tropopause temperature, deep convection, gravity waves, stratospheric aerosol,



and their relations are presented in Section 3. The data uncertainties and relation uncertainties between SICs and all parameters are discussed in Section 4. Conclusions are presented in Section 5.

## 2 Data and method

### 2.1 Tropopause data

The first tropopause (1st-TP) is defined as the lowest level at which the lapse rate decreases to 2° C/km or less, provided the average lapse rate between this level and all higher levels within 2 km does not exceed 2° C/km based on World Meteorological Organization definition (WMO, 1957). If the average lapse rate at any level and at all higher levels within one kilometer exceeds 3° C/km above the 1st-TP, the second tropopause (2nd-TP) is defined by the same criteria as the first tropopause. The cold point tropopause (CP-TP) is defined as the minimum in the vertical temperature profiles (Highwood and Hoskins, 1998). Unlike the CP-TP, which is more related to atmospheric activities mainly over tropics (Munchak and Pan, 2014; Pan et al., 2018), the first thermal tropopause is a globally applicable tropopause definition to identify the transition between the troposphere and stratosphere (Munchak and Pan, 2014; Xian and Homeyer, 2019). Therefore, thermal tropopause (1st-TP and 2nd-TP) are analyzed in this work to conduct the stratospheric ice clouds on a global scale.

Tropopause heights are derived from the fifth generation European Centre for Medium-Range Weather Forecasts' (ECMWF's) reanalysis - ERA5, which is produced using 4D-Var data assimilation and model forecasts in CY41R2 of the ECMWF Integrated Forecast System (IFS) (Hersbach et al., 2020). ERA5 provides hourly high-resolution data from 1979 to present with a horizontal grid resolution of 0.25° and 137 hybrid sigma/pressure levels vertically from the surface to 0.01 hPa. The vertical resolution of ERA5 data is about 300‑360 m around the first tropopause level at the altitude range from 9 to 18 km. In our study, the vertical resolution of tropopause height is improved after interpolating the ERA5 data to a much finer vertical grid with cubic spline interpolation method (Hoffmann, 2021a).

In previous studies, e. g., Homeyer et al. (2010); Pan and Munchak (2011); Zou et al. (2020) used a tropopause threshold of 500 m to identify stratospheric clouds. Because of the higher vertical resolution of ERA5, which is improved by a factor of 2 compared with ERA-Interim, 250 m is employed as a valid tropopause threshold for ERA5 to extract stratospheric ice clouds in this study.

### 2.2 CALIPSO observations of stratospheric ice clouds and stratospheric aerosols

The Cloud-Aerosol Lidar with Orthogonal Polarization (CALIOP), which is a dual-wavelength polarization-sensitive lidar instrument loaded on CALIPSO satellite, probes high-resolution vertical structure and properties of thin clouds and aerosols on a near-global scale since June 2006 (Winker et al., 2007, 2009). CALIPSO equatorial crossing times are at 01:30 local time (LT) for the descending orbit and 13:30 LT for ascending orbit sections. The vertical resolution of CALIPSO observations varies as a function of altitude, which is 60 m at a range from 8.2 to 20.2 km.





Ice cloud and aerosol are extracted from the Vertical Feature Mask data (CAL_LID_L2_VFMStandardV4) in this study. According to the cloud and aerosol subtype classifications determined by the CALIPSO's cloud-aerosol discrimination (CAD)

algorithm and the International Satellite Cloud Climatology Project (ISCCP) definitions, ice clouds in this work include cirrus clouds and deep convective clouds. Aerosols are dust, contaminated dust, and volcanic ash. Samples marked with high feature type quality are used to ensure high reliability of data. Considering that aerosols in the lower stratosphere are relatively long-lived and can contribute to the production of ice clouds, both day- and nighttime aerosols are included. As for ice clouds, only nighttime data are investigated due to their higher signal-to-noise ratios and detection sensitivity with extinction coefficients

down to $10^{-3}$ km$^{-1}$ (Getzewich et al., 2018; Gasparini et al., 2018).

The highest sample of cloud and aerosol in each profile are extracted to identify stratospheric ice clouds and aerosols. Only those with clouds/aerosols top heights (CTHs/ATHs) at least 250 m above the 1st-TP are defined as stratospheric ice clouds (SICs) and stratospheric aerosols (SAs). The filter criterion for polar stratospheric clouds (PSC) (Sassen et al., 2008), i. e., data are excluded if CTHs are higher than 12.0 km in areas with local winter latitude $\geq 60°$ N and $60°$ S, are utilized here to avoid

possible miscounting of PSC. This filter criterion doesn't apply to some low altitude PSCs. Due to large uncertainties of data over polar region, SICs detected at high latitudes will not be discussed in detail in this work. The occurrence frequency of SICs and SAs is defined as the ratio of SIC/SA detection numbers to total profile numbers in a given region.

## 2.3   AIRS observations of deep convection, gravity waves and SO$_2$

The Atmospheric Infrared Sounder (AIRS) (Aumann et al., 2003; Chahine et al., 2006) is carried by NASA's Aqua satellite.

AIRS has the same equatorial crossing time as CALIPSO at 01:30 LT for the descending orbit and at 13:30 LT for ascending orbit. It measures the thermal emissions of atmospheric constituents in the nadir and sublimb viewing geometry. There are 2378 spectral channels in total for AIRS infrared spectrometer with a spectral coverage of 3.74 to 4.61 μm, 6.20 to 8.22 μm and 8.8 to 15.4 μm. Over the full dynamic range from 190 K to 325 K, the absolute accuracy of each spectral channel is better than 3 % and noise is less than 0.2 K at 250 K scene temperature (Aumann et al., 2000).

Deep convection, gravity waves and sulfur dioxide (SO$_2$) emissions from volcanic eruptions can be retrieved from AIRS in wavebands at 8.1 μm, 4.3 μm and 7.3 μm, respectively (Aumann et al., 2003, 2006; Hoffmann and Alexander, 2010; Hoffmann et al., 2013, 2014a,b, 2016). Since a constant brightness temperature (BT) threshold for deep convection detection may lead to ambiguous results at different latitudes and seasons (Hoffmann et al., 2013), temperature differences between AIRS brightness temperatures (BT$_{AIRS}$) at 1231 cm$^{-1}$ (8.1 μm) and tropopause temperatures (T$_{TP}$) from ERA5 are used to detect deep

convection events (Zou et al., 2021). An offset of $+7$ K on top of the T$_{TP}$ was set as the threshold to include all possible deep convection events with cloud tops near or above the tropopause,

$$BT_{AIRS} - T_{TP} \leq 7\,K. \tag{1}$$

The choice of the temperature threshold determines the absolute values of the occurrence frequencies of the deep convection events, but it does not fundamentally affect the spatial and temporal patterns of deep convection (Zou et al., 2021). The term

"deep convection" here includes convection from storm systems and fronts, mesoscale convective systems, and mesoscale



convective complexes. Similar to Hoffmann et al. (2013), monthly mean brightness temperatures at mid- and high latitudes are applied to filter cases with low surface temperatures, which make it particularly difficult to detect convection. Observations are removed if monthly mean brightness temperatures are below 250 K over regions with latitude $> 25°$ N/S.

As the occurrence (coverage) of SICs associated with deep convection depends on the intensity, spatial extent, and duration of the deep convection, detection numbers of SICs and deep convection may not be the best indicator to elucidate their relations (Zou et al., 2021). Therefore, to effectively investigate the relation of SICs and deep convection, event frequency is defined in this work as the ratio of number of days in which deep convection or SICs ($\geq 1$ detection) occurs to the total number of days in a given time period over a given region. This event frequency evaluates how frequently deep convection and SICs occur rather than their coverage, and it is largely independent of intensity, spatial extent or the duration of the deep convection.

The mean variance of brightness temperatures in the 4.3 $\mu$m waveband is used to identify stratospheric gravity wave signals from AIRS observations (Hoffmann and Alexander, 2010; Hoffmann et al., 2013). Measurements of 42 AIRS channels from 2322.6 to 2345.9 $cm^{-1}$ and 2352.5 to 2366.9 $cm^{-1}$ are averaged to reduce noise and improve detection sensitivity of the AIRS gravity wave observations. This detrended and noise-corrected mean BT variance has the highest sensitivity at an altitude range of $30 - 40$ km (Hoffmann and Alexander, 2010; Hoffmann et al., 2013, 2018). However, it can also provide information on gravity waves in the lower stratosphere because gravity waves typically propagate upward from the tropospheric sources into the stratosphere. Instead of setting a variance threshold to identify gravity wave events (Hoffmann and Alexander, 2010; Zou et al., 2021), here we use mean BT variances directly as a measure of gravity wave activity. A higher mean BT variance indicates a larger amplitude of the gravity waves. Note that the BT variance is strongly dependent on both, the gravity wave sources and the background winds in the stratosphere, and its observation is intermittent in time, i. e., monthly or seasonal mean values can smooth its characteristics. It is also important to keep in mind that like most satellite instruments, AIRS is only capable of observing a specific part of the full stratospheric spectrum of gravity waves. Being a nadir instrument, AIRS is most sensitive to short horizontal and long vertical wavelength waves (Ern et al., 2017; Meyer et al., 2018).

As brightness temperature differences are an effective method to detect volcanic $SO_2$ from AIRS observations (Hoffmann et al., 2014b, 2016), spectral features of $SO_2$ at 1407.2 $cm^{-1}$ and 1371.5 $cm^{-1}$ are used to calculate the $SO_2$ Index (SI),

$$SI = BT(1407.2\,cm^{-1}) - BT(1371.5\,cm^{-1}). \qquad (2)$$

The SI represents the $SO_2$ column density from the mid troposphere to the stratosphere, where a high SI indicates a high $SO_2$ column density. The SI is most sensitive to $SO_2$ layers at an altitude range from 8 to 13 km and an $SI > 4$ K is most likely related to volcanic emissions (Hoffmann et al., 2014b). In this work, an SI threshold of 10 K is applied to identify relatively strong volcanic eruptions whose emissions may affect the lower stratosphere.





## 3 Results

### 3.1 Global lower stratospheric ice clouds

Ice clouds with cloud top heights at least 250 m above the first tropopause were defined as stratospheric ice clouds (SICs). Figure 1 a-d present the global distribution and mean occurrence frequency of SICs from 2007 to 2019 in December-January-February (DJF), March-April-May (MAM), June-July-August (JJA) and September-October-November (SON). Seasonally averaged occurrence frequencies of ice clouds as a function of altitude are shown in Fig. 1 e-h. Similar to the results of previous studies (Pan and Munchak, 2011; Zou et al., 2020; Dauhut et al., 2020), enhanced occurrences of SICs are observed at the tropical continents. The highest SIC frequencies over the tropics are detected in boreal winter (DJF) (~0.36) with the regional mean of ~0.15. The weakest signal of SICs over the tropics occurs in boreal summer (JJA), when the hotspots of SICs are shifted to the north of the equator over the Asian Monsoon and North American Monsoon. In the midlatitudes, more SICs are observed in the northern hemisphere over the northern Atlantic and Europe in DJF. In JJA, only the region over central North America presents relatively high SIC frequencies (0.08-0.12). In the southern hemisphere, SICs are observed continuously along mid- and high latitude belts in JJA. MAM and SON have similar features as DJF and JJA. In the vertical, ice clouds are mostly found at the tropopause region (± 500 m around the tropopause).

As defined in Section 2, the occurrence frequency is the ratio of ice cloud detections to the total number of profiles in a given region. As the occurrence (coverage) of SICs associated with deep convection depends not only on spatial extent but also on the intensity and duration of the deep convection, the event frequency is proposed in this work as the ratio of days with SIC or deep convection detection to the total number of days in a given time period to effectively investigate the relation between SICs and deep convection. The global mean event frequencies of SICs by season are presented in Appendix A. Generally, the mean event frequencies show similarities with the occurrence frequency of SICs in Figure 1.

The second tropopause is identified if the average lapse rate at any level and at all higher levels within one kilometer exceeds 3° C/km above the first tropopause. The existence of a second tropopause indicates a less stable temperature structure and double tropopauses may enhance deep convection to the lower stratosphere (Homeyer et al., 2014). The frequencies of SICs with cloud top heights more than 0.25 m above the first tropopause but below the second tropopause are shown in Fig. 2 a-d. SICs coinciding with double tropopauses are mostly observed in midlatitudes, e. g., over the northern Pacific Ocean, northern Atlantic near the United States, and Tibetan Plateau in DJF and MAM, over central North America and southern South America in JJA and SON. In the tropics, there are about 2 - 4 % of the SICs associated with double tropopauses, mainly located over the Maritime Continent in DJF, equatorial Africa in MAM, and the northeastern Indian Ocean in JJA and SON. The occurrence of double tropopauses in general greatly impacts the SICs' occurrences associated with double tropopauses. The patterns of SICs associated with double tropopauses in Fig. 2 a-d resemble the patterns of occurrence frequencies of double tropopauses in Peevey et al. (2012) and Schwartz et al. (2015). The presence of double tropopauses is associated with the subtropical jet stream, which enhances transport from the tropics to higher latitudes (Randel et al., 2007). In addition, Fig. 2 e-h show the fraction of SICs associated with double tropopauses to the total occurrence frequency of SICs. Up to 80-100 % of all SICs around a latitude band of 30° in both hemispheres during local winter and autumn are associated with double tropopauses. In





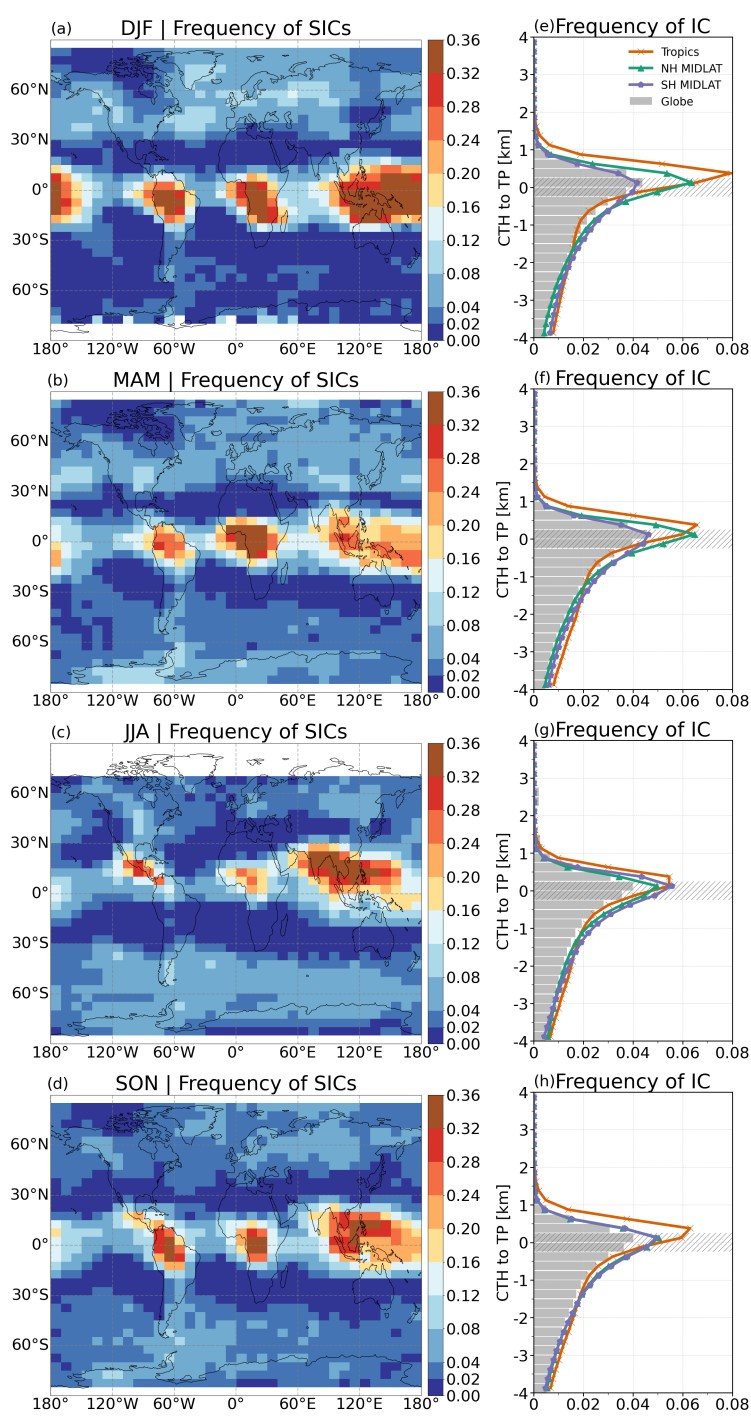

**Figure 1.** Occurrence frequencies of SICs on a $5° \times 10°$ (latitude × longitude) grid (a-d) and occurrence frequencies of ice clouds in the altitude range from $-4$ to $4\,\mathrm{km}$ with respect to the first thermal tropopause (e-h) in DJF, MAM, JJA and SON. The data are shown as zonal averages, globally, for the tropics ($20°\,\mathrm{S}$ - $20°\,\mathrm{N}$) and midlatitudes ($40°$ -$60°$).





the tropics over the Maritime Continent in DJF, equatorial Africa in MAM and northeastern Indian Ocean in SON less than

40 % of the SICs coincide with double tropopauses. Only over the northwestern Indian Ocean in JJA up to 60 % of the SICs are associated with double tropopauses. This indicates that double tropopauses have a non-negligible impact on the occurrence of SICs, especially in and around the subtropical jet stream. SICs associated with double tropopauses in the polar region may be due to misclassified PSCs and larger uncertainties in the thermal tropopause due to the relatively constant temperature profile in the polar winter UTLS. In this work, we focus our analyses mainly on the tropics and midlatitudes.

To investigate spatial and temporal variations of SICs, monthly averaged occurrence frequencies of SICs in $5°$ latitude bands from 2007 to 2019 are shown in Fig. 3. Seasonal cycles and inter-annual variability of SICs are observed in all latitude belts. In the tropics, the high SIC occurrence frequency varies with the seasonal motion of the Sun, with the highest frequency generally occurring at $20°$ S in boreal winter, then gradually moving to $20°$ N in boreal summer. High frequencies of SICs (>0.20) are found in $15°$ S - $5°$ N in November to March while relatively low values (0.16 - 0.24) are found at latitudes $10°$ N - $20°$ N in July

to September. In mid- and high latitudes, SICs are more often observed from December to May in the northern hemisphere and from May to October in the southern hemisphere. In the tropics there are several pronounced high SIC occurrence frequencies in December 2008 to February 2009, November 2010 to January 2011, December 2011, January 2013, February 2018 to April 2018, and November 2018 to January 2019. Some relatively low occurrence frequencies between $10°$ N - $20°$ N are visible in boreal summer in 2012, 2014 to 2017. There is a consistency in the spatial distribution of SICs over time, i. e., SICs in the

tropics follow the Intertropical Convergence Zone (ITCZ) and also follow an annual pattern in the midlatitudes. However, interannual changes are also evident. In the following we investigate possible explanations for the SICs' global patterns and their temporal variability.



**Figure 2.** The occurrence frequencies of SICs associated with the double tropopauses (a-d) where the cloud top heights are 0.25 km above the first tropopause and below the second tropopause. e-h) are the fraction of SICs associated with double tropopauses to total SICs.





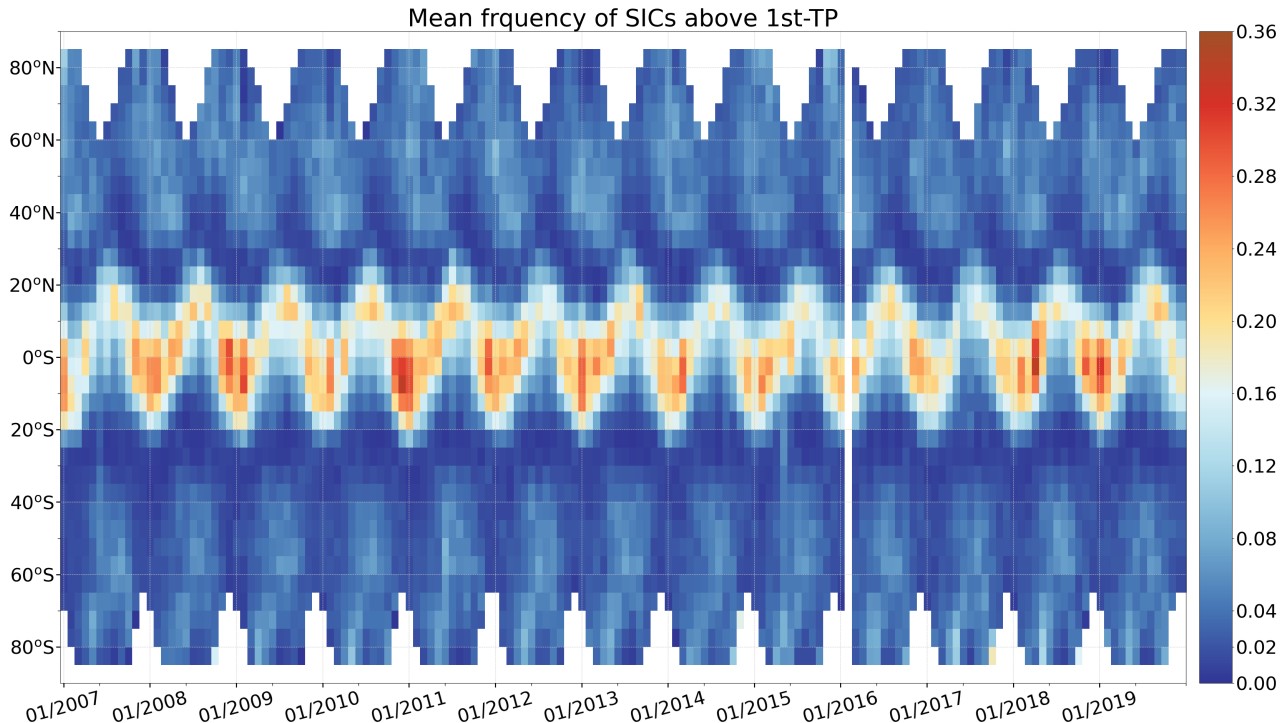

**Figure 3.** Monthly mean occurrence frequencies of SICs in latitudes band of $5°$ from 2007 to 2019.

## 3.2 Tropopause temperature and SICs

Tropopause temperature plays a vital role in influencing ice clouds and regulating water vapor in the lower stratosphere.
Low temperatures are more favorable for ice formation, and temperature normally has negative relation with cirrus cloud
frequency (Eguchi and Shiotani, 2004; Kim et al., 2016). To better understand the effects of tropopause temperature on the
global distribution and occurrence of SICs, seasonal mean first tropopause temperature and SICs frequencies are presented in
Fig. 4. Low tropopause temperature are characteristic in the tropics, where large-scale updrafts, convection, and waves cause
its cooling. As already noted by Chae and Sherwood (2007) that tropopause temperatures over tropics are colder in boreal
winter than summer, we can find higher occurrence frequencies of SICs over tropics in DJF than in JJA (Fig. 4). In general,
regions with low tropopause temperature are co-located with a high occurrence frequency of SICs.

To further investigate their relation, Spearman correlation coefficients between monthly averaged tropopause temperatures,
and occurrence frequencies of SICs from 2007 to 2019 for each grid cell (in $5°$ latitude $\times 10°$ longitude) are shown in Fig. 5a.
Only grid boxes with SIC frequencies larger than 0.02, with more than 80 (156 months in total) data points in each grid box
and correlation coefficients significance at 99 % level are presented here. The tropopause temperature is negatively correlated
with SIC frequency, especially in the SIC hotspots in the tropics, where r-values are generally $< -0.6$ and even reach up to
$< -0.8$ in several grid boxes in tropical South America, at the western coast of equatorial Africa, and northern Australia.





Negative correlations are also observed over mid- and high latitudes ($> 40°$). r-values$< -0.6$ are detected over Greenland, Iceland, and northwestern Europe. However, positive correlation coefficients are observed in several grid boxes in the North

American Monsoon, North African Monsoon, Indian Monsoon, and Western North Pacific Monsoon. The occurrence of SICs over tropical continents shows a significant dependence on tropopause temperature.

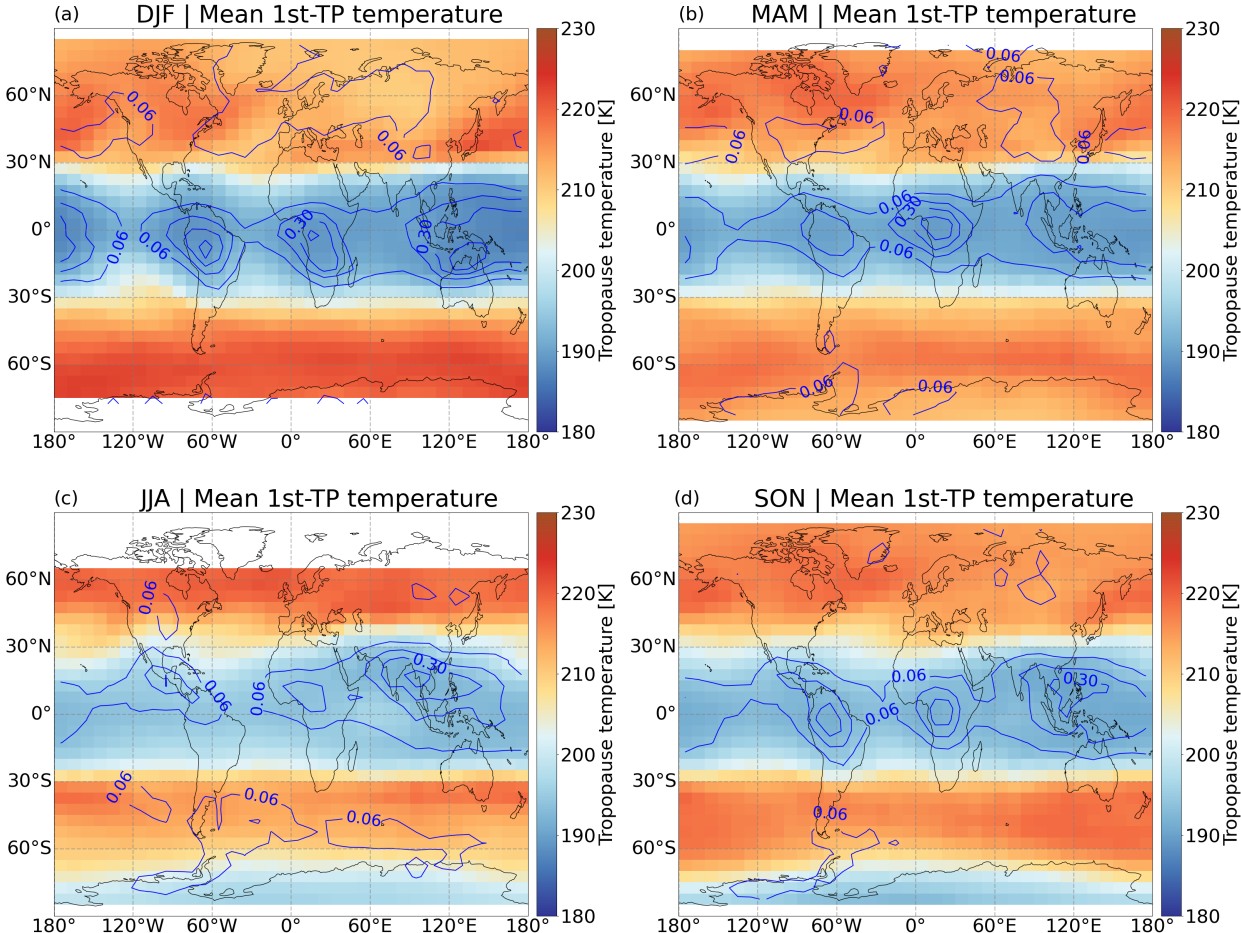

**Figure 4.** Seasonal mean first tropopause temperature derived from ERA5 (color boxes) and occurrence frequencies of SICs derived from CALIPSO (contour lines with the interval of 0.12).

Time series of monthly anomalies of SIC frequencies and tropopause temperatures at different latitude bands ($5°$ for each band) are shown in Fig. 6a and b. The monthly anomalies of SIC frequency and mean temperature for each grid box were computed as the difference between the monthly zonal mean values and the interannual mean of the monthly zonal mean





**Figure 5.** The Spearman correlation coefficients of SIC frequency with the first tropopause temperature, deep convection frequency, gravity waves, and stratospheric aerosol frequency. Only grid boxes with SIC frequency $> 0.02$, $\geq 80$ (in 156 months) total data points in each grid box, and at 99 % significance level are presented.

values,

$$For \; \theta_i \in [\theta - 5°, \theta], \; where \; \theta = [-80°, -75°, ..., 80°, 85°];$$

$$For \; m \in month \; [1, 2, 3, ..., 10, 11, 12];$$

$$D_T(\theta_i, m, y) = D(\theta_i, m, y) - \frac{1}{N} \sum_{y=2007}^{2019} D(\theta_i, m, y) \tag{3}$$

where $\theta_i$ is the latitude band, y $\in$ year [2007, 2008, ..., 2018, 2019], N is the number of years, which is 13; D($\theta_i$,m,y) is data
of SIC frequency or tropopause temperature, and D$_T$($\theta_i$,m,y) is the anomaly value.





As stated above, SIC frequencies and tropopause temperatures are generally negatively correlated. We expected anomalies of SIC frequencies and tropopause temperature to have opposite features in time series. Significant anomalies of SIC frequency are visible over the tropics (Fig. 6a). Anomalies of SIC frequencies and tropopause temperature overall show contrary features, such as February 2007 to July 2007, January 2008 to Jun 2008, October 2012 to Jun 2013, and July 2016 to Jun 2017. However, some remarkable positive anomalies of SIC frequencies between November 2010 to January 2011 coincide with positive temperature anomalies. Exceptions like these may be due to enhanced stratospheric aerosols (Noh et al., 2017; Chouza et al., 2020) or downward propagating of QBO (Feng and Lin, 2019; Tegtmeier et al., 2020b). Overall, tropopause temperatures are negatively correlated with the occurrence of SICs spatially and temporally, especially over tropical continents. In comparison, some positive correlations over monsoon regions and in the tropics at certain times need further investigation.



**Figure 6.** Monthly anomalies of SIC frequency derived from CALIPSO and the mean first tropopause temperature derived from ERA5 from 2007 to 2019.





### 3.3 Deep convection and SICs

Deep convection, which can inject water vapor and ice particles into the lower stratosphere and hence provides a source of humidity for in-situ nucleation above anvil tops, is closely related to the occurrence of SICs. As deep convection is the primary factor related to the occurrence of SICs over the Great Plains in storm season (Zou et al., 2021), here we investigate the correlation between deep convection and SICs on a global scale. To remove possible uncertainties related to the intensity, spatial extent and duration of deep convection, the event frequency of nighttime deep convection is investigated in this work (Fig. 7). As stated in Sect. 2, the event frequency is the ratio of the number of days with deep convection to the total number of days in one month, whereas the occurrence frequency is the ratio of profiles with deep convection to the total profile number in a specific grid box. The occurrence frequency of deep convection is shown in Appendix B. The patterns of event frequency and occurrence frequency are similar for deep convection in Fig. 7 and Fig. B1 and the seasonal patterns of event frequency are overall similar to results shown in Hoffmann et al. (2013). However, event frequencies are much higher than the occurrence frequencies and the results in Hoffmann et al. (2013) due to different computing methods and detection thresholds. In the tropics the highest event frequencies follow the ITCZ and are the strongest over the continents and southeastern Asia. In the midlatitudes, the highest event frequencies are over the oceans and southern South America in DJF and the highest frequencies are observed over the continents in JJA (Fig. 7).

Correlation coefficients between monthly averaged event frequency of SIC and deep convection from 2007 to 2019 are shown in Fig. 5 b. The SIC event frequencies are generally positively correlated with deep convection. High correlations are found over Central America, tropical South America, southern Asia, maritime continent and northern Australia, southern Africa and Madagascar with the highest coefficient $> 0.8$, which demonstrates that the occurrence of SICs over those regions is highly correlated with the occurrence of deep convection. As a colder tropopause aligns with the higher frequency of tropopause-penetrating convection over the tropics (Gettelman et al., 2002), we conclude that the interaction between tropopause temperature and deep convection both associate with the occurrence of SICs based on the co-location of large correlation coefficients in Fig. 5 a and b.

To further investigate the effects of deep convection, we analyzed the fraction of SICs related to deep convection (Fig. 8), which is defined as the ratio of the number of days with co-occurrence of SICs and deep convection ($N_{(SICs \cap DC)}$) to the number of days with occurrence of SICs ($N_{(SICs)}$) in each grid box. Observations at the same local time (LT) for SICs and deep convection, which is named as 0 local time difference (0 LTD), are presented in Fig. 8. In DJF (Fig. 8 a) more than 50 % of the SICs are correlated with deep convection over Argentina and southern Brazil, eastern Tibetan Plateau (with maximum fraction of 80-90 %), northern Pacific Ocean (maximally 70-80 %) and maritime continent. In JJA (Fig. 8 c), the highest correlations between SIC and deep convection are observed over the Great Plains (maximally 70-80 %), Central America (with the highest fraction of 90-100 %), central Africa (about 50-60 %), eastern and southern Asia, Europe and Western Pacific Ocean, and over a latitudinal band along 30°S-45°S (40-80 %). During boreal summer, more than 40 % of SICs over the northern hemisphere continents and southern hemisphere oceans are correlated with deep convection. In MAM (Fig. 8 b), regions with the largest correlations are similar to JJA but with lower statistics. In SON, regions with the highest correlations between SICs and deep





**Figure 7.** Seasonal event frequency of deep convection (DC) derived from AIRS during 2007-2019.

convection are similar to JJA and DJF (Fig. 8 d). The pattern of a high fraction is similar to positive vertical velocity within

cirrus clouds for corresponding months in Barahona et al. (2017, Fig.06). Overall, the influence of deep convection on the

occurrence of SICs follows the ITCZ in the tropics. Most high values are observed in JJA. SICs detected over the Great Plains,

North American Monsoon and the Asian Monsoon in JJA and over northern Pacific in DJF are mainly attributed to deep

convection.

    As the lifetime of TTL cirrus may be as long as 12-24 h (Jensen et al., 2011), we also analyzed the correlation with deep

convection observed by AIRS measurements 12 hours (-12h LTD) and 24 hours (-24h LTD) before (Appendix. C). The left

column of Fig. C1 shows fractions of SICs related to deep convection, which are detected at 0 LTD and -12h LTD (DC at -

12h ∪ 0h LTD) to SICs, and the right column (Fig. C1) are fractions of SICs related to deep convection detected at 0 LTD, -12h

LTD and -24h LTD (DC at -24h ∪ -12h ∪ 0 LTD) in different seasons. We find that fractions of SICs related to deep convection

generally increase by 10 % when one more time detection is included. Therefore, more SIC occurrence can be traced back to





deep convection if the lifetime of SICs is taken into account. However, since the lifetime of SICs is unavailable in our dataset, we mainly focus on the deep convection detected simultaneously with SICs.



**Figure 8.** The fraction of SICs associated with deep convection at same local time (0 LTD) (color box). Occurrence frequency of SICs are shown in red contours with the interval of 0.12.

## 3.4 Gravity waves and SICs

Gravity waves are crucial factors locally affecting the pressure, temperature, and vertical velocity of an air parcel. As cold phase and cooling effects of gravity waves have a significant influence on cirrus cloud occurrence (Chang and L'Ecuyer, 2020;

Ansmann et al., 2018), it is essential to investigate the relation between gravity waves and the occurrence of SICs. Mean variances of brightness temperatures (BT) at 4.3 $\mu$m from AIRS observations are applied to identify gravity wave amplitudes. Due to the wind filtering and visibility effects, gravity waves in AIRS measurements are not significantly observed in tropics (Hoffmann et al., 2013). In JJA, hotspots of large amplitude waves (mean BT variance > 0.1 K$^2$) are observed at mid- and high



latitudes in the southern hemisphere, especially over Patagonia, Drake Passage, and the Antarctic Peninsula. In the Northern
Hemisphere, high variance is found over southern and southeastern Asia, the Great Plains, Florida, and northern Africa in
Fig. 9c. In DJF (Fig. 9 a), high variance (>0.1 K$^2$) is observed over the northern Atlantic, eastern Canada and the United States,
Europe and the mean variances of all regions north of 40°N, except for the northern Pacific Ocean, are greater than >0.03
K$^2$. In the southern Hemisphere, several gravity wave hotspots are detected over southern Africa and Madagascar, northern
Australia and coral sea, and southern Brazil. In MAM, gravity waves are observed mainly over the southern hemisphere, with a
similar pattern to that in JJA, but with weaker signals, and southern Greenland (Fig. 9 b). Similar patterns with weaker signals
to DJF are observed in SON over the northern hemisphere, and an intense center is detected over Patagonia, Drake Passage
and the Antarctic Peninsula in this time (Fig. 9 d). Regions with high mean BT variance for each season are in agreement with
hotspots of SICs (Fig. 1) and deep convection (Fig. 7).

The correlation coefficients between the monthly averaged frequency of SICs and mean BT variance are shown in Fig. 5c.
Positive correlation coefficients are discovered in most regions of the world. High correlations with correlation coefficients
> 0.8 are found over southern Asia, northern Australia, southern Africa and Madagascar, central Pacific Ocean, and Central
America. This indicates that the amplitude of gravity waves has a significant impact on the occurrence of SICs. In addition,
high correlation regions of mean BT variance overlapped with high correlations with deep convection over Central America,
tropical South America, southern Asia, southern Africa, and northern Australia (Fig. 5b and c), which provides information for
high correlation of deep convection and gravity waves. However, positive correlation of SICs and gravity waves and negative
correlations of SICs and deep convection are found in high latitudes, which indicates the important role of gravity waves on
the occurrence of SICs in high latitudes. All in all, we conclude that gravity waves and deep convection are important factors
influencing the occurrence of SICs, interactively and respectively.

### 3.5   Stratospheric aerosols and SICs

The time series of SIC frequency anomalies (Fig. 10a) and stratospheric aerosol anomalies (Fig. 10b) are presented together
with significant volcanic eruptions derived from AIRS SO$_2$ measurements, where occurrences of SO$_2$ Index (SI)> 10 K are
marked as triangles. The largest positive anomalies of SICs are found in the tropics at 0° - 20° S in November 2010 to January
2011, 5° N - 20° N in June and July 2011, 5° S - 5° N in June and July 2015, 15° S - 20° N in April and May 2018, and 5° N -
25° N in September to November 2019. No reasonable explanation could be obtained from relation analyses of SIC frequency
with deep convection, tropopause temperature and gravity waves. However, we noticed that those high SIC frequencies coincide
with volcanic eruptions of Merapi (7.5° S) in November 2010, Nabro (13° N) in June 2011, Wolf (0°) in May 2015, Ambae
(15.4° S) in April 2018, and Ulawun (5° S) in August 2019, respectively (Global Volcanism Program, 2013; Hoffmann, 2021b).
At mid-latitudes the most pronounced positive anomalies in SIC frequency correlate with the ash rich volcanic eruptions of
Kasatochi (August 2008, 52° N), Puyehue-Cordón Caulle (June 2011, 41° S) Calbuco in April/May 2015 (41° S) and Raikoke
(June 2019, 48° N) (compare with AIRS ash and SO$_2$ index Hoffmann, 2021b). High SIC anomalies coincide to a large extent
with volcanic eruptions and hence, suggest a potential influence of aerosol on the CALIPSO ice cloud product.







**Figure 9.** The mean brightness temperature variance at 4.3 $\mu$m from AIRS measurements, which indicates the amplitude of gravity waves. Occurrence frequency of SICs are shown in blue contours with the interval of 0.12.

Daily frequencies of SAs and frequency anomalies of SICs over tropics and midlatitudes in different years are selected as examples to demonstrate their relations in Fig. 11. In Figure 11a, high SA frequency and high SIC frequency anomalies are found on day 160-180 in 2011 at southern midlatitudes, where an SI>10 was found approximately 10-20 days before.

Similar patterns we found between day 90 to 120 in the tropics in 2018 (Fig. 11c) and between day 170 and 210 at northern midlatitudes in 2019 (Fig. 11d). In these cases, stratospheric aerosol injected by volcanic eruptions, such as Puyehue-Cordón Caulle in 2011 (Klüser et al., 2013), Ambae in 2018 (Malinina et al., 2021), and Raikoke in 2019 (Kloss et al., 2021), show strong relationships with large positive SIC frequency anomalies. The enhanced SA and SIC frequency anomaly between days 220 and 240 in 2017 at northern midlatitudes (Fig. 11b) is related to wildfires over the United States and Canada in August and

September 2017 that greatly increased stratospheric aerosol load (Ansmann et al., 2018; Selimovic et al., 2019).





**Figure 10.** Monthly anomalies of SIC frequency and stratospheric aerosol frequency from CALIPSO observations from 2007 to 2019. Blue triangles indicate volcanic events, identified by an SI > 10 K derived from AIRS observations.

Long-term correlations between monthly mean frequency of SICs and stratospheric aerosol are presented in Fig. 5d. Positive correlations are generally observed over the world. High coefficients (> 0.6) are located over southern Asia, northern Australia, southern Africa and Madagascar, the central Pacific Ocean, and Central America. Those are known regions with large-scale upwelling and tropopause-penetrating convection, which may lead to the vertical transport of aerosol from the troposphere to the stratosphere. The likely cloud seeding by Kasatochi volcanic aerosols (Campbell et al., 2012) demonstrate a close relation of atmospheric aerosols and occurrence of cirrus clouds. Whereas, strong inverse correlations were observed between stratospheric sulfur aerosol and cirrus cloud reflectance in Friberg et al. (2015). The positive correlations would also be affected by the classification method for ice and aerosol in CALIOP and the possible misclassification of them (Reverdy et al., 2012).






**Figure 11.** Daily frequency of stratospheric aerosols (SA) and daily anomaly of SIC frequency over tropic and midlatitudes from CALIPSO measurements in a) 2011, b) 2017, c)2018, d) 2019. Volcanic eruptions (SI > 10 K from AIRS measurements) are shown as green triangles. Blanks are missing data or filtered abnormal data that are three times greater than the standard deviation of regional mean frequency at that day.

## 3.6 Regional analyses

Relations between SICs and tropopause temperature, deep convection, gravity waves, stratospheric aerosol on a global scale were investigated above. All factors play important roles in the occurrence of SICs, but the importance varies seasonally from





region to region. Therefore, in this section selected regions are studied in detail. The regions shown in Fig. 12 were selected based on hotspots of SIC frequencies, i, e., Indo-Pacific Warm Pool (IPWP, 15°S-15°N, 90°E-180°E), Asian Monsoon (AM, 5°N-35°N, 50°E-120°E), southern South America (SSA, 30°S-60°S, 10°W-80°W), North American Monsoon (NAM, 0°-30°N, 60°W-120°W), tropical South America (TSA, 20°S-10°N, 40°W-100°W), equatorial Africa (EA, 15°S-15°N, 10°W-40°E) and northern Atlantic (NA, 40°N-75°N, 50°W-30°E).

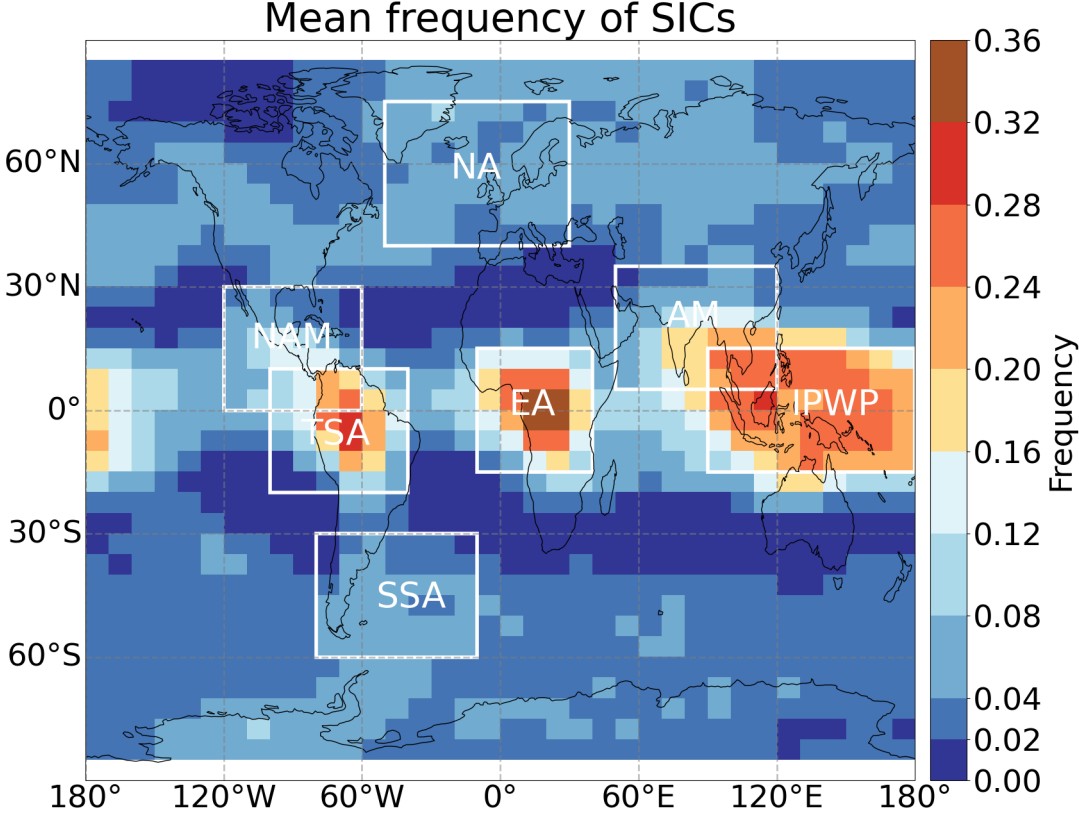

**Figure 12.** The annual mean SICs frequency. White boxes are selected regions for further analysis.

Monthly first tropopause temperature (1st-TPT), stratospheric aerosol frequency (SAs), gravity waves (GW, mean BT variance), monthly mean zonal wind over Singapore (1° N, 104° E) at a pressure level of 70 hPa, representing the Quasi-Biennial Oscillation (QBO) of the winds in the equatorial stratosphere (QBO 70hPa) and sea surface temperature over Niño 3.4 region (5° N-5° S, 170° W-120° W) (Niño3.4 SST) were analyzed along with anomalies of SIC frequency as well as anomalies of SIC and DC event frequencies over different regions. Correlation coefficients between SIC frequency and all other parameters are listed in Tab. 1. Averaged Spear-man correlation coefficients were obtained by re-sampling all data 50 times using the Boot-strap method (Efron and Tibshirani, 1985). Three regions of monthly anomalies of all parameters are selected and presented in Fig. 13-Fig. 15. The anomalies were calculated by subtracting all month averaged data from monthly values. All data in February 2016 were excluded due to the absence of CALIPSO measurements.





**Table 1.** Mean correlation coefficients of SIC frequency with first tropopause temperature (1st-TPT), event frequency of deep convection frequency (eDC), stratospheric aerosol frequency (SAs), mean BT variance (GW), QBO 70hPa, Niño3.4 SST and their standard deviations from a bootstrapping analysis.

| Regions | 1st-TPT | eDC | SAs | GW | QBO 70hPa | Niño3.4 SST |
|---|---|---|---|---|---|---|
| **IPWP** (15°S-15°N, 90°E-180°E) | -0.67±0.05 | 0.40±0.07 | 0.54±0.05 | 0.37±0.09 | -0.21±0.06 | -0.50±0.06 |
| TSA (20°S-10°N, 40°W-100°W) | -0.81±0.04 | 0.18±0.09 | 0.60±0.05 | 0.47±0.06 | -0.06±0.09 | -0.20±0.08 |
| EA (15°S-15°N, 10°W-40°E) | -0.71±0.05 | 0.22±0.08 | 0.62±0.05 | 0.06±0.08 | -0.09±0.08 | -0.12±0.08 |
| **AM** (5°N-35°N, 50°E-120°E) | -0.55±0.06 | 0.92±0.01 | 0.52±0.05 | 0.71±0.04 | -0.04±0.07 | 0.11±0.07 |
| NAM (0°-30°N, 60°W-120°W) | 0.15±0.09 | 0.89±0.02 | 0.51±0.07 | 0.54±0.05 | 0.00±0.07 | 0.01±0.08 |
| **SSA** (30°S-60°S, 10°W-80°W) | -0.78±0.04 | 0.61±0.05 | -0.08±0.08 | 0.74±0.03 | 0.04±0.08 | 0.11±0.07 |
| NA (40°N-75°N, 50°W-30°E) | -0.83±0.04 | 0.60±0.05 | 0.04±0.08 | 0.66±0.05 | 0.06±0.08 | -0.10±0.09 |

Over Indo-Pacific Warm Pool (IPWP), the variation of SICs is strongly correlated with the 1st-TPT (-0.67±0.05), SAs (0.54±0.05) and Niño3.4 SST (-0.50±0.06) (Tab. 1). From Fig. 13, we find SIC frequencies are higher in winter and spring (NDJF) and lower in summer and autumn (JJAS), following the ITCZ. Seasonal fluctuations in SIC frequencies are similar to SA frequencies and gravity waves, but are inversely correlated with 1st-TPT. Some high frequencies of SICs are found in November to next year February in 2007, 2008, 2009, 2018 and December 2011, January 2013. Cold tropopause, high loading of stratospheric aerosols, large gravity wave amplitude all show high relationships with those high SICs, i. e., the coldest tropopause and cold phase of ENSO in November 2007 to the following February, peak gravity wave in January 2013, high SA frequencies in November 2018 to the following February. During November 2009 and 2015 to the following February, low SIC frequencies coincide with the warm phase of ENSO, when tropopause is warm and less convective cloud ice is produced over the Western Pacific (Avery et al., 2017). Fluctuations of tropopause temperature are critical factors in the seasonal cycle of SIC frequency, while SA, GW and atmospheric turbulence all affect variations of SICs over IPWP.

Over the Asian Monsoon region, anomalies of SIC frequencies are high in JJA and low in DJF. Seasonal cycles of SIC and deep convection event frequencies coincide well in time (Fig. 14b), following the ITCZ, with a correlation coefficient of 0.92±0.01 indicating the crucial role of deep convection on the occurrence of SICs over the Asian Monsoon. Similar to temporal variation of SIC and deep convection frequency, gravity waves also show a high correlation with the occurrence of SICs (r=0.71±0.04). In the summers of 2007, 2010, 2012-2014, 2017 and 2019, high SIC frequencies are influenced by deep convection and gravity waves. The peak SIC frequency in July 2011 correlates with enhanced stratospheric aerosol after the volcanic eruption of Nabro in June 2011. Two highest SIC frequency anomalies in June-September 2010 and 2013 are related to QBO's easterly phase, La Niña phenomenon, enhanced deep convection and gravity waves. Deep convection and gravity waves are two main factors related to the occurrence and variability of SICs over the Asian Monsoon region.

Over southern South America, positive anomalies of SIC frequencies are detected in JJA along with a cold tropopause (Fig. 15). Tropopause temperature is an important factor influencing the frequency of SICs with the correlation of -0.78±0.04. Gravity wave events and deep convection have a consistent seasonal cycle with SIC frequency with correlations of 0.74±0.03





and 0.61±0.05, respectively. The effect of stratospheric aerosols is limited in 2011 and 2019. High SAs frequencies are prob-
455 ably related to the volcanic eruptions of Puyehue-Cordón Caulle (June 2011), Ulawun (June-August 2019) and Ubina (July
2019). Orographically generated gravity waves (Hoffmann et al., 2013) and gravity wave induced temperature perturbations
may regulate the occurrence of SICs over southern South America, i. e., high SIC frequencies are highly correlated with high
wave amplitude in 2010, 2012, 2014, 2015, 2017 and 2018 and relatively low SIC frequencies are consistent with relatively
low BT variance in 2009, 2013 and 2016.

According to Tab. 1, variations of SIC frequencies over tropical South America and equatorial Africa have high correlations
with tropopause temperature and stratospheric aerosols, which is similar to IPWP. The highest correlations of SIC frequencies
with deep convection and gravity waves are observed over the Asian Monsoon and the North American Monsoon. Over
the northern Atlantic, gravity waves and tropopause temperatures show the strongest correlation with the variability of SIC
frequencies, which is similar to southern South Africa. As the highest correlations of processes with SIC frequencies over
465 the regions of tropical South America and equatorial Africa, North America Monsoon and northern Atlantic are similar to the
above analyzed three regions, the detailed information on those four regions are presented in Sect. D. On the whole, tropopause
temperature, deep convection, gravity wave, stratospheric aerosol and atmospheric turbulence are all important factors related
to the occurrence and variability of SICs. However, leading factors differ over regions.



**Figure 13.** Monthly anomalies of SIC frequency, tropopause temperature(1st-TPT), event frequencies of SIC and deep convection, stratospheric aerosol frequency, gravity waves, QBO at 70hPa and Niño3.4 SST over Indo-Pacific Warm Pool (IPWP 15° S-15° N, 90° E-180°) from 2007 to 2019.



**Figure 14.** Monthly anomalies of SIC frequency, tropopause temperature(1st-TPT), event frequencies of SIC and deep convection, stratospheric aerosol frequency, gravity waves, QBO at 70hPa and Niño3.4 SST over Asian Monsoon (5° N-35° N, 50° E-120° E) from 2007 to 2019.



**Figure 15.** Monthly anomalies of SIC frequency, tropopause temperature(1st-TPT), event frequencies of SIC and deep convection, stratospheric aerosol frequency, and gravity waves over southern South America (30° S-60° S, 10° W-80° W) from 2007 to 2019.





## 4  Discussion

### 4.1  Tropopause threshold and SICs identification

In this work, a tropopause threshold of 250 m is applied to identify stratospheric ice clouds and stratospheric aerosols (Sect. 2.1). While 500 m is an acceptable threshold for tropopause uncertainty (Homeyer et al., 2010; Pan and Munchak, 2011) and it is solidly applied to ERA-Interim data (Zou et al., 2020, 2021), 250 m is a reasonable tropopause threshold for ERA5 data due to its two times higher vertical resolution than in ERA-Interim. Moreover, the lapse rate tropopause height difference between ERA5 and radiosonde data (Global Navigation Satellite System-Radio Occultation) of less than 200 m in the tropics (Tegtmeier et al., 2020a, Fig. 08) further supports the 250 m uncertainty of ERA5 tropopauses. In the tropics the SIC occurrence frequencies using ERA5 tropopauses with a threshold of 250 m above the tropopause is identical with the SIC occurrence frequency we found using ERA-interim reanalyses and a threshold of 500 m above the tropopause (Zou et al., 2020). Although one would expect a higher SIC occurrence frequency when using a smaller distance to the tropopause, the results remain identical. The major reason for this finding is that the ERA5 tropopauses in the tropics are in average 100 to 150 m higher than the ERA-interim tropopauses (Hoffmann and Spang, 2021, Fig6a, at 0 °) and hence, compensate most of the effect of a lower distance to the tropopause. At midlatitudes, however, about three times more SICs are detected in this study using ERA5 tropopauses (Fig. 1) compared to Zou et al. (2020, Fig.3) using ERA-interim tropopauses. The statistical analysis of ERA-interim and ERA5 tropopause heights shows that depending on season and hemisphere, the mean midlatitude tropopause in ERA5 is between 100 m lower to 80 m higher than the ERA-interim tropopause (Hoffmann and Spang, 2021, Fig6a, at 45°). Hence, the ERA5 tropopause at midlatitudes remains the approximately same, lowering of the threshold distance to the tropopause results in more cloud detections, as one would expect.

### 4.2  Deep convection frequency uncertainties

Clouds are labeled as deep convection if the cloud top brightness temperature is close to the tropopause temperature with an offset of 7 K from the tropopause temperature. The temperature threshold plays an important role in defining deep convection (Zou et al., 2021). A high threshold will include more tropospheric clouds, and a low threshold may miss some deep overshooting. Even though the numbers of deep convection detection may vary slightly, the global pattern of deep convection is robust across different temperature thresholds.

The occurrence frequency of deep convection is usually defined as the coverage of deep convection in a region, which is the ratio of deep convection detections to total observations. However, limitations of occurrence frequency will stick out when we investigate its relation with the occurrence of SICs since the intensity, spatial extent and duration of deep convection are important features affecting the occurrence of SICs. Therefore, event frequency is proposed in this work to investigate their relations, which is defined as the number of days with deep convection/SIC detection to the total number of days in a specific time period and grid box.

The event frequency can effectively demonstrate the relations of SICs and deep convection. Even though there are large quantitative differences between event frequency and occurrence frequency of deep convection, the global patterns of event


frequency and occurrence frequency are matched (Fig. 7 and Fig. B1). The event frequencies are larger than 40 % over Northern Pacific in DJF, over Central America, the Great Plains, Maritime continent in JJA, but the occurrence frequencies are about 3 %. Event frequency can get rid of the effects of the intensity, spatial extent and duration of deep convection. For example, signals of occurrence frequency (Fig. B1) over the tropics are much weaker than that in event frequency (Fig. 7). It means deep convection over tropics happens as frequent as in midlatitudes on time scale, but the spatial extents are smaller and the intensities are stronger in the tropics than in midlatitudes.

### 4.3 Stratospheric aerosols and SICs uncertainties

Stratospheric aerosols (dust, contaminated dust and volcanic ash) were extracted from CALIPSO measurements to investigate their correlation with SICs. High correlation coefficients of SICs and SAs (Fig. 5 d) and some high SIC frequencies co-occurring immediately with or with 1-2 month lag after large volcanic eruptions or wildfires (Fig. 10 and Fig. 11) indicate potential effects of volcanic aerosol and biomass burning on the observation of SICs with CALIPSO.

Despite the recent improvement in the CALIOP aerosol and cloud discrimination (Liu et al., 2019) we investigated potential aerosol cloud misclassifications further by comparing the SIC anomalies of CALIOP and MIPAS measurements in the over-lapping measurement period between January 2007 and April 2012 (Fig. 16). As MIPAS is an IR limb emission instrument and its algorithm for classification between ice, volcanic ash, and sulfate aerosol is entirely different to CALIPSO, as it relies on spectral signatures (Griessbach et al., 2014, 2016), we assumed that it does not necessarily show the same anomalies. In the southern hemisphere midlatitudes one (June 2011, $40° - 65°$ S) out of two positive SIC anomalies between 2007 and 2012 in the CALIOP data coincides with the volcanic plume after the eruption of Puyehue-Cordón Caulle in June 2011. In the MIPAS data, this anomaly is not visible. The eruption of Puyehue-Cordón Caulle is known to have injected significant amounts of vol-canic ash (Klüser et al., 2013; Hoffmann et al., 2014a). Moreover, Klüser et al. (2013) show that "the ash plume is transported very close to and potentially partly within or beneath ice clouds". In such a case the CALIOP "cloud fringe amelioration" algo-rithm might rather classify these detections as ice clouds instead of aerosol (Liu et al., 2019). Moreover, Liu et al. (2019) point out that the aerosol cloud classification for this volcanic plume was particularly challenging due to the dense and depolarizing aerosol.

In the northern hemisphere midlatitudes also one significant positive SIC anomaly (August to October 2008, $45° - 80°$ N) in the CALIOP data coincides with the volcanic plume after the eruption of Kasatochi in June 2008. In the MIPAS, this anomaly is not visible, but starting from November 2008 a positive anomaly is visible. The Kasatochi eruption is known to have mainly injected $SO_2$ (1.21 Tg) and some ash (0.31 Tg) (e.g. Prata et al., 2010). However, the volcanic aerosol plume following the eruption of the Sarychev volcano in June 2009, which injected somewhat less $SO_2$ (1 Tg) (Clarisse et al., 2012) and a slightly smaller fraction of ash (Andersson et al., 2013), does not coincide with a positive SIC anomaly. The major difference between both plumes is that the Kasatochi plume was distributed around the tropopause at altitudes between 9.1 -- 13.7 km (Corradini et al., 2010), whereas the Sarychev plume was distributed over a larger altitude range and reached higher into the stratosphere with plume heights between 8.5 and 17.5 km (e.g. Doeringer et al., 2012). Especially the higher plume height makes it less likely to be interpreted as an ice cloud in the lowermost stratosphere.





In the tropics the two strongest anomalies are correlated with the volcanic eruptions of Merapi in November 2010 (November 2010 to January 2011, SH tropics) and Nabro in June 2011 (May to July 2011, NH tropics) (Fig. 10 and Fig. 16). In both cases the MIPAS data also shows a positive, but weaker, anomaly. Although volcanic aerosol is known to induce ice cloud formation and the MIPAS data also shows some (not discussed) significant anomalies, which are not present in the CALIOP data, we 540 consider the positive event-related anomalies rather a misclassification in CALIPSO data.

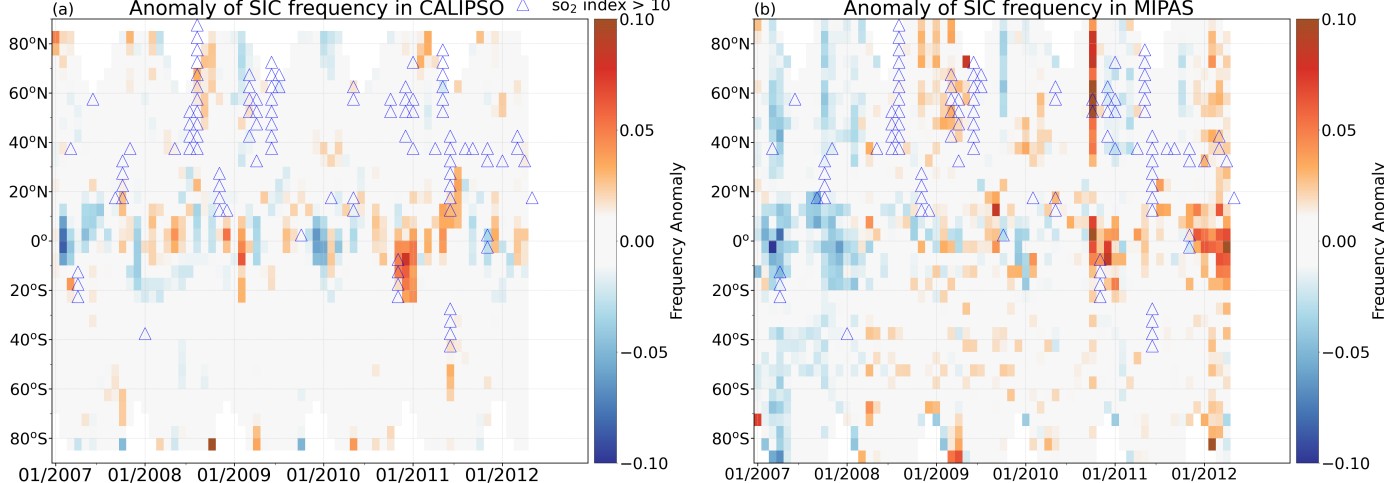

**Figure 16.** Monthly anomalies of SIC frequencies from CALIPSO (a) and MIPAS (b) measurements. Blue triangles are data with SI > 10 K derived from AIRS observations.

### 4.4 Correlations of SICs with related factors

Tropopause temperature, deep convection, stratospheric aerosol and gravity waves are important factors related to the occurrence of SICs over different regions. For example, the correlation coefficients between SIC and deep convection frequencies are up to 0.92 over the Asian Monsoon and 0.89 over the North American Monsoon; correlation coefficients of SIC frequency 545 and tropopause temperature are -0.78 and -0.67 over southern South Africa and the Indo-Pacific-Warm-Pool (Table 1). Long-term fluctuations of all parameters presented in Fig. 13-Fig. 15 greatly reveal their relations. However, correlation coefficients may be affected by linear trends of factors. Long-term trends for all parameters were calculated (Table 2) and detrending was processed for all phenomena. Re-calculated correlation coefficients after detrending are shown in Table 3. Detrended data was calculated as the difference between the observed data and predicted values from a regression model. A general decrease in 550 tropopause temperature in all regions is in line with decreasing trend of tropical tropopause temperature reported by Randel et al. (2006) and Tegtmeier et al. (2020a). In contrast, a general increase in stratospheric aerosol frequencies is observed in all regions. After detrending, correlation coefficients of SIC with other factors changed slightly, but the correlation coefficients with SA frequencies generally increased. Correlation coefficients are up to 0.73±0.05, 0.79±0.05 and 0.82±0.05 over IPWP, TSA and EA, respectively, suggesting an enhanced correlation between stratospheric aerosol and the occurrence of SICs.



**Table 2.** Slopes of monthly SIC frequency (pp/year), 1st-TPT (K/year), event frequencies of SICs (pp/year) and deep convection (eSICs, eDC) (pp/year), SAs (pp/year), GW (0.01*K$^2$/year), QBO 70hPa (m/s/year), Niño3.4 SST (K/year) during 2007-2019

| Regions | SICs | 1st-TPT | eSICs | eDC | SAs | GW | QBO 70hPa | Niño3.4 SST |
|---|---|---|---|---|---|---|---|---|
| IPWP (15°S-15°N, 90°E-180°E) | -0.21 | -0.21 | -0.07 | -0.02 | 0.03** | 0.00 | -0.08 | 0.09 |
| TSA (20°S-10°N, 40°W-100°W) | -0.32* | -0.19 | -0.17 | -0.04 | 0.04** | -0.01 | - | - |
| EA (15°S-15°N, 10°W-40°E) | -0.22 | -0.23 | -0.10 | -0.09 | 0.03** | -0.01 | - | - |
| AM (5°N-35°N, 50°E-120°E) | -0.08 | -0.24 | -0.10 | -0.04 | 0.00 | 0.00 | - | - |
| NAM (0°-30°N, 60°W-120°W) | -0.11 | -0.17 | -0.03 | 0.11 | 0.01** | 0.00 | - | - |
| SSA (30°S-60°S, 10°W-80°W) | 0.05 | -0.31 | 0.07 | -0.07 | 0.04** | 0.15 | - | - |
| NA (40°N-75°N, 50°W-30°E) | -0.08 | -0.28 | -0.09 | 0.03 | 0.00 | 0.09 | - | - |

Note: * data are validated at 95 % confidence level, ** data are validated at 99 % confidence level.

pp/year is the percentage point per year.

Slope values of QBO 70hPa and Niño3.4 SST are all the same over regions, marked as '-'.

**Table 3.** Correlation coefficients and standard deviations after detrending

| Regions | 1st-TPT | eDC | SAs | GW | QBO 70hPa | Niño3.4 SST |
|---|---|---|---|---|---|---|
| IPWP (15°S-15°N, 90°E-180°E) | -0.58±0.05 | 0.43±0.07 | 0.73±0.05 | 0.36±0.09 | -0.22±0.06 | -0.47±0.06 |
| TSA (20°S-10°N, 40°W-100°W) | -0.74±0.04 | 0.20±0.09 | 0.79±0.05 | 0.44±0.06 | -0.05±0.09 | -0.14±0.08 |
| EA (15°S-15°N, 10°W-40°E) | -0.66±0.05 | 0.23±0.08 | 0.82±0.05 | 0.04±0.08 | -0.08±0.08 | -0.10±0.08 |
| AM (5°N-35°N, 50°E-120°E) | -0.56±0.06 | 0.92±0.01 | 0.57±0.05 | 0.71±0.04 | -0.06±0.07 | -0.10±0.07 |
| NAM (0°-30°N, 60°W-120°W) | 0.17±0.09 | 0.89±0.02 | 0.67±0.07 | 0.52±0.05 | -0.01±0.07 | 0.04±0.08 |
| SSA (30°S-60°S, 10°W-80°W) | -0.72±0.04 | 0.63±0.05 | -0.19±0.08 | 0.71±0.03 | 0.04±0.08 | 0.07±0.07 |
| NA (40°N-75°N, 50°W-30°E) | -0.79±0.04 | 0.61±0.05 | 0.01±0.08 | 0.64±0.05 | 0.06±0.08 | -0.05±0.09 |

## 5   Conclusions


Stratospheric ice clouds (SICs) have an important impact on the water vapor balance and radiative budget in the UTLS, but knowledge of their formation and variation is still limited. In this study, we analyzed 13 years (2007-2019) of satellite observations by CALIPSO and AIRS together with data from ERA5 reanalyses to investigate the long-term variability of SICs and potential effects of tropopause temperature, deep convection, gravity waves, and stratospheric aerosol on the occurrence of

SICs.

A SIC is defined as an ice cloud with a cloud top height 0.25 km above the first thermal tropopause derived from ERA5 temperatures in this study. In agreement with results in Pan and Munchak (2011) and Zou et al. (2020), SICs are mainly detected over tropical continents and more SICs are observed in local winter in both hemispheres. The occurrence of SICs between the first and second thermal tropopause suggests a vertical instability of the UTLS. SICs associated with the double

tropopauses are mostly located at midlatitudes (between $25° - 60°$) in winter time. Nearly 80-100 % of the SICs associated with



double tropopauses are observed around 30°N/S during local winter and autumn, which is related to the polar-ward isentropic transport and mixing of water vapor in the lowermost stratosphere (Randel et al., 2007; Peevey et al., 2012).

Monthly 5° latitudinal band mean SIC occurrence frequencies from 2007 to 2019 were analyzed to present the seasonal cycle and inter-annual variability of SICs. Lower tropopause temperatures are usually co-located with a higher frequency of SICs.
A generally inverse correlation between tropopause temperature and SIC occurrence frequency is detected on the global scale (Fig. 5 a), especially in the tropics, where the largest negative coefficient is $< -0.8$ over tropical South America, equatorial Africa and northern Australia.

Hotspots of deep convection and gravity waves are similarly located over the tropical continents, western and northern Pacific, North America (Great Plains), northern Atlantic, Central America, Argentina and southern Brazil. Gravity waves have
stronger signals at mid- and high latitudes in local winter. The same patterns were found for the correlation between SICs and deep convection, SICs and gravity waves (Fig. 5 b,c). Positive high correlations ($>0.8$) are found in Central America, tropical South America, southern Asian, maritime continent and northern Australia, southern Africa and Madagascar. However, the high correlation between deep convection and gravity waves themselves (Hoffmann et al., 2013) leads to an overlap of their effects on SIC occurrence. Therefore, the fractions of the SICs related to deep convection (Fig. 8) were calculated to quantify
the impact of deep convection as much as possible. Our results show that more than half of the SICs are related to deep convection over the northern Pacific, maritime continent, Mediterranean and Black Sea, Argentina and southern Brazil in DJF. In JJA, up to 80 % SICs are related to deep convection over the Great Plains and Central America.

The global relation between SICs and stratospheric aerosol was also analyzed as the coincidence of high SIC frequencies and volcanic eruptions and wildfires were detected in some cases (Fig. 11). Worldwide positive correlations between SICs and SAs
are observed in this work. High correlation coefficients ($> 0.6$) are mostly detected over southern Asian, northern Australia, southern Africa and Madagascar, the central Pacific Ocean and Central America. However, this positive correlation might be affected by the misclassification of aerosol and ice in CALIPSO (Reverdy et al., 2012).

Above we presented high correlation of SIC occurrence frequency with double tropauses, tropopause temperature, deep convection, gravity waves and stratospheric aerosol on a global scale. However, inherent correlations of the above listed factors,
such as high correlation between deep convection and gravity waves (Hoffmann et al., 2013), variation of temperature excited by deep convection and gravity waves (Gettelman et al., 2002) and sudden increase of stratospheric aerosol transported by vertical upwelling and wave propagation, make it difficult to clarify their impacts on SIC occurrence. Small regions were selected to specify the impacts of different factors. Results show that tropopause temperature and stratospheric aerosol show the highest correlation with the high frequency of SICs over the Indo-Pacific Warm Pool, tropical South America and equatorial
Africa. Deep convection and gravity waves are highly correlated with the occurrence of SICs in the Asian Monsoon and the North American Monsoon region. Gravity waves and tropopause temperature have high correlation coefficients with SIC occurrence frequency in southern South America and the northern Atlantic.

Based on satellite observations from CALIPSO and AIRS, we investigated the global distribution and long-term variability of stratospheric ice clouds and assessed its relations with tropopause temperature, deep convection, gravity waves and strato-
spheric aerosol. Generally, positive correlations are observed between SICs and deep convection, gravity waves, stratospheric





aerosol, and inverse correlations are detected between SICs and tropopause temperature. However, the specific role of different factors varies over regions. To further explore the formation mechanisms and precisely elucidate the origin of SICs, Lagrangian modelling and microphysical simulations are required in future studies.

*Data availability.* Convection and gravity wave data from AIRS used in this study are available at https://www.re3data.org/repository/

r3d100012430 (last access: 3 December 2020) (Hoffmann, 2020). ERA5 tropopause data are available at https://www.re3data.org/repository/ r3d100013201 (last access: 25 November 2021) (Hoffmann, 2021a). The AIRS volcanic data are available at https://datapub.fz-juelich.de/ slcs/airs/volcanoes (last access: 01 July 2021) (Hoffmann, 2021b). Monthly mean zonal wind over Singapore are obtained from https://www. geo.fu-berlin.de/en/met/ag/strat/produkte/qbo/index.html and SST data are obtained from https://www.ncdc.noaa.gov/teleconnections/enso/ indicators/sst/. Cirrus cloud top heights from CALIPSO are available upon request from the contact author, Ling Zou (l.zou@fz-juelich.de;

cheryl_zou@whu.edu.cn).





## Appendix A: Event frequency of SICs

Figure. A1 shows the seasonal event frequencies of SICs. Global features are similar to occurrence frequencies in Fig. 1 that hotspots of SICs are located in tropical continents. However, event frequencies are lower than occurrence frequencies in tropics but higher than occurrence frequencies in midlatitudes. High latitudes will not be discussed in detail as high frequencies over there may relate to the occurrence of PSC. From Fig. A1 and Fig. 1, we can find that SICs are more often detected over tropics than in midlatitudes on time scale and the horizontal extent of SICs over tropics is much wider than that in midlatitudes.

**Figure A1.** Event frequencies of SICs on a $5° \times 10°$ (latitude $\times$ longitude) grid box from CALIPSO measurements during 2007-2019.





## Appendix B: Occurrence frequency of deep convection

Occurrence frequency of deep convection in AIRS are presented in Fig. B1. In DJF, high frequencies of deep convection are found in northern Pacific, Alaska, western Canada, northern Atlantic close to the United States, eastern and western sided of Tibetan plateau, Argentina and southern Brazil, northern Australia, Mediterranean and black sea region. In JJA, hotspots of deep convection are located at central North America (Great Plains), Central America, central Africa, southern Asia and Western Pacific Ocean, southern Brazil and latitudinal band in 30°S-45°S. MAM and SON are intermediate seasons which have similar regions of high deep convection frequency as DJF and JJA. The seasonal patterns and values of deep convection frequency are overall similar to results shown in Hoffmann et al. (2013). Similar patterns can be found both in event frequency and occurrence frequency of deep convection, but signals in Fig. 7 are much stronger than that in Fig. B1, which indicates that high frequencies of deep convection on time scale but relatively small spatial extent of it on coverage.

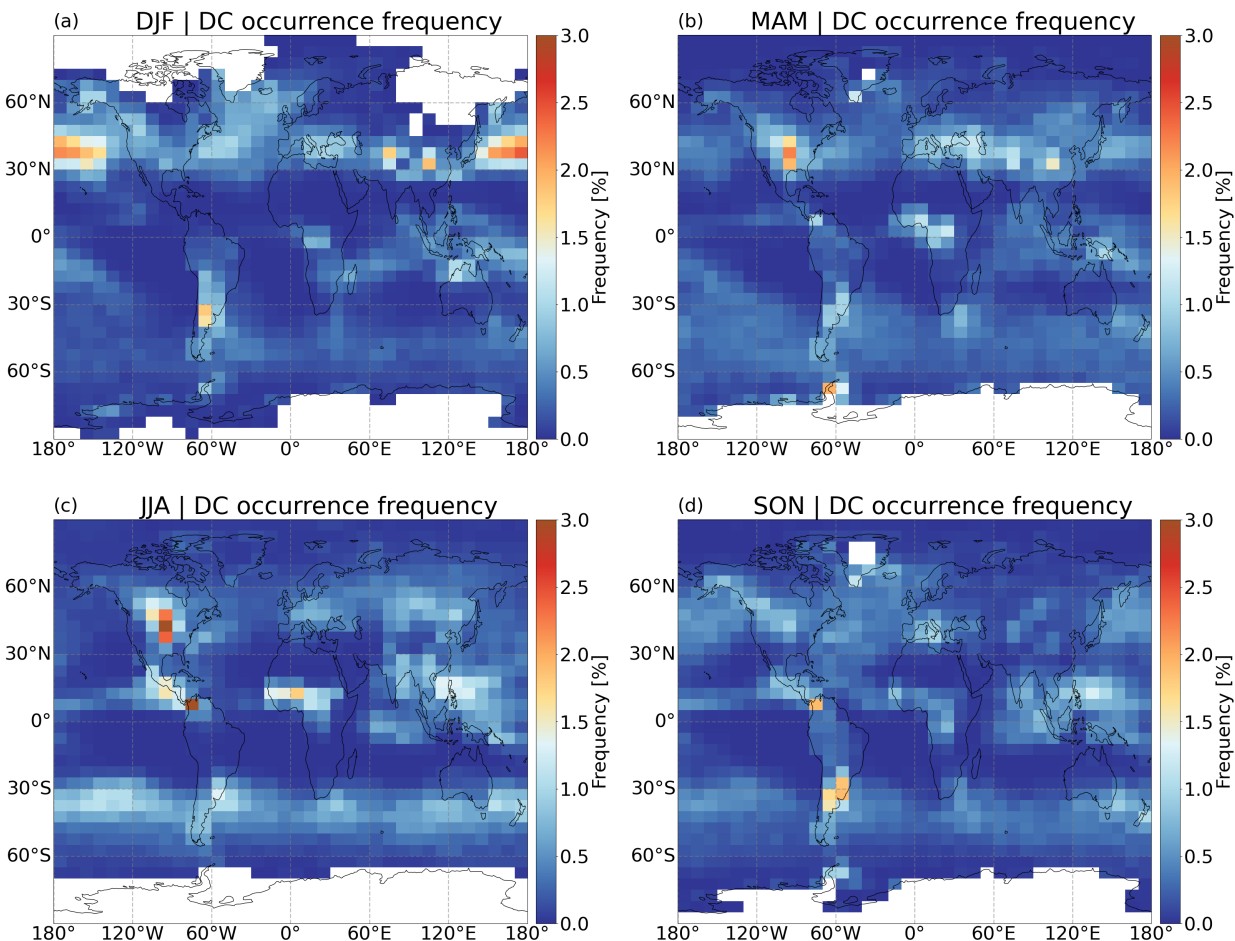

**Figure B1.** Seasonal mean occurrence frequency of deep convection derived from AIRS measurements during 2007-2019.



**Appendix C: Fraction of SICs related to deep convection**

Considering possible effects of deep convection occurred before detection of SICs, deep convection from AIRS observations detected at 12 hours (-12h LTD) and 24 hours (-24h LTD) before are analyzed in Fig. C1. Left column shows fractions of
SICs related to deep convection, which are detected at 0 LTD and -12h LTD (DC at -12h∪0h LTD) to SICs, and right column (Fig. C1) are fractions of SICs related to deep convection detected at both 0 LTD, -12h LTD and -24h LTD (DC at -24h∪-12h∪0 LTD) in difference seasons. By comparing results in Fig. 8 and Fig. C1, we find that about 10 % more SICs are related to the deep convection when one more time detection included. It means more SIC occurrence can be traced back to deep convection if the lifetime of SICs would be considered.



**Figure C1.** The fraction of SICs related to deep convection with deep convection observed by AIRS measurements at 12 hours (-12h LTD) and 24 hours (-24h LTD) before, the occurrence frequency of SICs are shown in red contours with the interval of 0.12.





## Appendix D: Other four regions

Over tropical South America, strong correlations of SIC frequencies are found with the tropopause temperature and SAs (Fig. D1). High SIC frequencies in 2008-2011 are co-located with cold tropopause temperature, positive SA frequency anomalies and high brightness temperature variances. However, the relation of SICs and deep convection is low with the correlation coefficient only of 0.18±0.09.

Over equatorial Africa, the strongest correlation of SIC frequencies is detected with the tropopause temperature with a correlation coefficient of -0.71±0.05 (Fig. D2). The correlation coefficient between SIC frequencies and SA frequencies is also high, and we can find an increasing trend of SA frequencies from 2007 to 2019. The increased stratospheric aerosols in 2011, 2012, 2013, 2018 and 2019 are consistent with the high SIC frequencies. The large amplitudes of gravity waves in 2011-2013 show high correlations with the high frequencies of SICs. Like the other two tropical regions (IPWP and TSA), event frequencies of deep convection over EA present low correlations with SIC occurrence.

Over the North American Monsoon region (Fig. D3), seasonal fluctuations of deep convection are highly consistent with SIC frequency with a correlation of 0.89±0.02. Gravity waves are generally in line with SIC frequencies with a correlation of 0.54±0.05. High SIC frequency from June to September 2015 is strongly dependent on stratospheric aerosols, which may be produced by a volcanic eruption (Wolf (0°) in May 2015). Deep convection is highly associated with the occurrence and variability of SICs over the North American Monsoon region.

Over the North Atlantic (Fig. D4), tropopause temperature and gravity waves have the strongest correlations with the occurrence of SICs. The low tropopause temperature and high brightness temperature variance align with high SIC frequency anomalies in winter. The event frequencies of deep convection and SICs are also positively correlated over this region. The relation of SICs and SAs is relatively small, and the high SA frequencies may be due to misclassified PSCs over high latitudes.

*Author contributions.* LZ, LH, SG and RS conceived the study design. LH provided the AIRS data and the ERA5 tropopause data. SG provided the MIPAS data. LZ processed the CALIPSO data and compiled all results. LZ wrote the manuscript with contributions from all authors.

*Competing interests.* The authors declare that they have no conflict of interest.

*Acknowledgements.* This work was supported by the German Research Foundation (DFG) through the AeroTrac project under the grant ID: DFG HO5102/1-1. We gratefully acknowledge the computing time granted on the supercomputers JURECA and JUWELS at Forschungszentrum Jülich. CALIPSO data are obtained from the NASA Langley Research Center Atmospheric Science Data Center. The AIRS data were distributed by the NASA Goddard Earth Sciences Data Information and Services Center. The ERA5 data were obtained from the European Centre for Medium-Range Weather Forecasts. The MIPAS data were provided by the European Space Agency.

**Figure D1.** Monthly anomalies of SIC frequency, tropopause temperature(1st-TPT), event frequencies of SIC and deep convection, stratospheric aerosol frequency, and gravity waves over Tropical South America ($20°$ S -$10°$ N, $40°$ W-$100°$ W), from 2007 to 2019.

**Figure D2.** Monthly anomalies of SIC frequency, tropopause temperature(1st-TPT), event frequencies of SIC and deep convection, stratospheric aerosol frequency, and gravity waves over equatorial Africa (15° S -15° N, 10° W-40° E), from 2007 to 2019.



**Figure D3.** Monthly anomalies of SIC frequency, tropopause temperature(1st-TPT), event frequencies of SIC and deep convection, stratospheric aerosol frequency, and gravity waves over North American Monsoon (0° -30° N, 60° W-120° W), from 2007 to 2019.



**Figure D4.** Monthly anomalies of SIC frequency, tropopause temperature(1st-TPT), event frequencies of SIC and deep convection, stratospheric aerosol frequency, and gravity waves over North Atlantic (40° N -75° N, 50° W-30° E), from 2007 to 2019.



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
