# Peer review of "A global view on stratospheric ice clouds: assessment of processes related to their occurrence based on satellite observations"

_Atmospheric Chemistry and Physics, 2021_

## Author Comment (AC1)

**Reviewer #2**

**General Comments ∗ 1**

There is a lot of really interesting material here, but it seems somewhat unfocused and in places a little confusing. The manuscript is way too long for the average reader, I suggest tightening it up quite a bit and or consider writing two papers.

**Answer:** Thank you very much for your time and effort spending on this manuscript and for your helpful comments and suggestions. We have reorganized the structure of manuscript.

**General Comments ∗ 2**

SIC in the stratosphere might suggest moistening (as the authors state), but interest in that process is mostly associated with convection penetrating the tropical tropopause. In this paper, the extra-tropical and tropical SIC are sort of lumped together.
For example, SIC moistening in the stratospheric extra-tropics is unimportant since air is moving downward there and will soon exit the stratosphere. Whereas SIC in the tropics is very important since air is moving upward and SIC could be changing the water vapor budget. I guess my point is to wade through 34 pages of material here, the motivation could be clearer.

**Answer:** Most SICs are observed in the tropics which is of interest for the stratospheric moistening. Meanwhile, the SICs detected in the extra-tropics are also important to understand the physical processes of their formation mechanisms and water vapor transport in the lowermost stratosphere, for example, the convection at midlatitudes (Zou et al., 2021), isentropic transport from TTL (Spang et al., 2015), and the distribution of water vapor in the UTLS is associated with radiative effects that impact the global surface temperature (Riese et al., 2012).

**General Comments ∗ 3**

I think the correlation with gravity waves is really part of the correlation with tropopause temperatures and isn't a separate effect. In Fig. 5 you see that the correlation maps (5a) and (5b) are nearly identical. As noted below, the cold temperatures are due to a number of things and gravity waves will be one of them, I am not sure how you separate them in a correlation sense, but I am pretty sure they overlap.

**Answer:** Yes, tropopause temperatures, gravity waves and deep convection have strong correlations. In our study we could not separate their contributions to the occurrence of SICs. In the revised manuscript, the correlations between SIC frequencies and them are presented separately, but we added a discussion of their inherent correlations in Sect. 3.7.

'The occurrence of SICs has a general negative correlation with tropopause temperature, while SICs have positive correlations with UTLS clouds, gravity waves and stratospheric aerosols. The highest negative and positive correlations are mostly observed over the tropical continents and the western Pacific with correlation coefficients of $< -0.8$ between SICs and LRT1-T and $> 0.8$ between SICs and UTLS clouds, gravity waves, and stratospheric aerosols. High positive correlations are also found over the Asian Monsoon and the North American Monsoon regions between SICs and UTLS clouds, gravity waves, and aerosol. While the LRT1-T shows a general negative correlation, there are strong positive correlations over central America and the Caribbean Sea, Philippines and South Chinese Sea, and the Tibetan Plateau to the Caspian Sea. The highest correlation coefficients are as large as 0.8-1.0 in the North American Monsoon region, even for LRT1-T. In the Asian Monsoon region, negative correlations are detected over the Tibetan plateau, but positive correlations are seen over southern Asia and India between SICs and LRT1-T. High correlation coefficients imply the important role of tropopause temperature, UTLS clouds, gravity waves and stratospheric aerosols for the occurrence of SICs. However, overlapping high correlation coefficients indicate also strong connections between the tropopause temperature, UTLS clouds, gravity waves, and stratospheric aerosols themselves.

To further investigate the source of SICs, the highest and second-highest correlation coefficients between SICs and all processes for each grid box are shown in Fig. 11. Over the tropical continents, the highest correlation coefficients of SICs relate to tropopause temperature. The highest correlation coefficients are found between UTLS clouds and SICs in the monsoon domains in the latitude range between 15° and 30°, e.g., the North American Monsoon, the Asian Monsoon, the South African Monsoon regions and the La Plata basin. In the central United States, tropopause temperature and UTLS clouds have the highest correlations with SICs. Over Patagonia and the Drake Passage, tropopause temperature and gravtiy waves have the highest correlation with the occurrence of SICs. In the latitude range between 45° and 60°, the strongest correlations are found between SICs and tropopause temperature and gravity waves. However, the second-highest correlation coefficients of SICs are related to stratospheric aerosols, UTLS clouds, and gravity waves over the tropical continents, the North American Monsoon and the Asian Monsoon regions. The rather similar correlation coefficients of SICs with all processes indicate high correlations between all processes themselves.

For all processes, increased tropopause-penetrating convection may result in a cooler tropopause across the tropics (Gettelman et al., 2002). Gravity waves and wave breaking will locally cause a colder temperature in the atmosphere and air cooling (Dinh et al., 2016). High correlations were found between deep convection and gravity waves (Hoffmann et al., 2013), and vertical motion of air will transport aerosols into the stratosphere (Bourassa et al., 2012). The inherent correlations between all processes may help to explain the positive correlations between SICs and LRT1-T in the North American Monsoon and the Asian Monsoon regions. Even if the tropopause temperature is warm, UTLS clouds, gravity waves, and stratospheric aerosol could all contribute to the high occurrence frequency of SICs. For example, Fu et al. (2006) discovered that deep convection in the Asian Monsoon injected more ice and water vapor into the stratosphere with warmer tropopause temperatures. However, their strong correlation also makes it challenging to disentangle all processes' effects on the occurrence of SICs. '

**General Comments ∗ 4**

The SIC correlation with aerosols is interesting, but aerosols are associated with fires as well as volcanoes. The CALIPSO measurements are a direct measure, I don't see how SO2 adds anything to this analysis. You should also mention more clearly why you are correlating with aerosols. Basically, aerosols provide CN and thus clouds form at lower RH so cloud formation should be approximately correlated with aerosols if the environment is at saturation. Volcanic eruptions and fires contain a lot of water so there might be other things going on. I would argue that this whole analysis might be a separate and interesting paper. BTW, OMPS-LP also produces an excellent aerosol data set that is independent of CALIPSO.

**Answer:** Indeed, stratospheric aerosols from CALIPSO include aerosols associated with volcanic eruptions as well as fires. The $SO_2$ data was not investigated in detail, but rather used as a proxy for strong volcanic eruptions to indicate stratospheric aerosol that originated from volcanic eruptions or not.

The correlation with stratospheric aerosol is shown, because we realized that SIC anomalies (Fig. 8) coincide with enhanced volcanic aerosol. We fully agree that this finding deserves a more detailed investigation. However, although volcanic injections also contain enhanced water vapour and condensation/ice nuclei that were shown to lead to cloud formation in the troposphere, we cannot fully rule out classification artefacts in the CALIPSO data set, especially for ash-rich volcanic plumes. Using other aerosol data sets, such as the suggested OMPS-LP, would be very helpful. Yet, this was not the scope of our study, so we focused on showing the global correlation and discussed the potential implications and uncertainties in the discussion section.

**General Comments ∗ 5**

 One question I was left with is: Do SIC occurrences well above the tropopause differ from those that are near the tropopause. Your statistics in Figure 1 suggest occurrences up to about 1 km above, but are the outliers very different? Those clouds have more of a potential of moistening the stratosphere.

**Answer:** Based on Figure 1 in the manuscript, we found that most ice clouds are observed around the tropopause. As you pointed out, there are approximately 1 % of ice clouds with tops 1 km above the tropopause. We didn't discuss those ice clouds separately from all SICs. However, for your interest, we present those ice clouds with cloud tops ≥ 1 km above the tropopause here in Fig. 1. Those ice clouds are mainly observed over the tropical continents in all seasons and also over central North America in JJA. They would be highly related to the deep convection (Fig. 5 in the manuscript).

[Figure]

Figure 1: Occurrence frequencies of ice clouds with cloud tops ≥ 1 km above the tropopause.

**General Comments ∗ 6**

 Please focus Abstract on salient points, it is way too long.

**Answer:** Thank your for your suggestion. We have revised the abstract in the manuscript.

'Ice clouds play an important role in regulating the water vapor and influencing the radiative budget in the atmosphere. This study investigates stratospheric ice clouds (SICs) based on the Cloud-Aerosol Lidar and Infrared Pathfinder Satellite Observations (CALIPSO). Tropopause temperature, double tropopauses, clouds in the upper troposphere and lower stratosphere (UTLS), gravity waves and stratospheric aerosols, were analyzed to investigate their relationships with the occurrence and variability of SICs in the tropics and at midlatitudes.

We found that SICs with cloud top heights of 0.25 km above the first lapse rate tropopause are mainly detected in the tropics. Monthly time series of SICs from 2007 to 2019 show that high frequencies of SICs follow the Intertropical Convergence Zone (ITCZ) over time in the tropics and that SICs vary inter-annually at different latitudes. Results show that SICs associated with double tropopauses, which are

related to poleward isentropic transport, are mostly found at midlatitudes. More than 80 % of the SICs around 30°N/S are associated with double tropopauses.

Correlation coefficient and long-term anomaly analyses of SICs and all the other processes indicate that the occurrence and variability of SICs are mainly associated with the tropopause temperature in the tropics. UTLS clouds have the highest correlations with SICs in the monsoon regions and the central United States. Tropopause temperature and gravity waves are mostly related to SICs at midlatitudes, especially over Patagonia and the Drake Passage. However, besides the highest correlation coefficients, the cold tropopause temperature, the occurrence of double tropopauses, high stratospheric aerosol loading, frequent UTLS clouds and gravity waves all have high correlations with the SICs. The occurrence and variability of SICs demonstrate a strong dependence on various processes, both locally and temporally.

The overlapping and similar correlation coefficients between SICs and all processes indicate strong associations between all processes themselves. Due to their high inherent correlations, it is challenging to disentangle and evaluate their contributions to the occurrence of SICs on a global scale. However, the correlation coefficient analyses between SICs and all processes and high associations between all processes observed in this study help us better understand the sources of SICs on a global scale.'

**Specific Comments:**

1. Ln 38 net radiative heating? You need to be careful here. High thick cirrus will produce surface heating by blocking IR cooling to space. Do you mean in situ cooling?

We have rephrased this sentence. 'Ice clouds in the UTLS region produce net radiative heating by trapping outgoing longwave radiation (Zhou et al., 2014; Lolli et al., 2018).'

2. Ln 43 Paragraph starting line 43, I presume you are not including Polar Stratospheric Clouds in this discussion. What does 'high altitude' mean? Above the tropopause?

The 'high altitude' indicates clouds observed in the upper troposphere and lower stratosphere. Yes, polar stratospheric clouds are not discussed in our work. We have revised this sentence in the manuscript. 'Global occurrence of ice clouds in the UTLS is about 20 – 40 % over the world (Liou, 1986; Wylie et al., 1994, 2005).'

3. Ln 59 Sentence starting with 'With ... ' makes no sense to me.

We have rephrased this sentence. 'It is critical to have a better understanding of the potential formation mechanisms and maintenance of ice clouds in the UTLS. '

4. Ln 67 Awkward English. 'Nucleation of ice crystals occurs in the presence of cold temperatures.'

This paragraph has been revised.

5. Overall comment on ln 34-119. On one hand this is a very thorough review of the literature, but its goal is occasionally illusive. Here is what I got from 80 lines of text: Ice clouds form in cold temperatures Cold temperatures are due to dynamics. Ice clouds are also injected by convection Aerosols impact cloud formation. I suggest that the reader might benefit from a reorganization of this material along these points.

Thank you, we have reorganized the whole section.

6. Ln 148 Spline interpolation can produce a new minimum temperature which is colder than the adjacent model levels, and the actual location of the minimum between the two model levels is unknown. Given the reliance of this study on the exact location of the tropopause, it seems appropriate to have more extended discussion which occurs later in the paper. Perhaps you could move that here. More explicit discussion of the uncertainty in the tropopause height might me appropriate.

Thank you. We have extended the discussion on the tropopause uncertainty in the revised manuscript in Sect. 2.1 and Sect. 4.1. We have used the same tropopause data as Hoffmann and Spang (2022). They found that, after using spline interpolation, the uncertainty of the first lapse rate tropopause height for ERA5 is in ±200 m at different latitudes compared to the US High Vertical Resolution Radiosonde Data (HVRRD) data and the coarser-resolution Global Positioning System (GPS) data. Therefore, we consider the 250 m tropopause threshold used in our study is reasonable.

'Tegtmeier et al. (2020a) found that LRT1 height differences between ERA5 and Global Navigation

Satellite System-Radio Occultation observations are less than 200 m in the tropics. Based on US High Vertical Resolution Radiosonde Data (HVRRD) data and coarser-resolution Global Positioning System (GPS) data, Hoffmann and Spang (2022) also showed that the uncertainty of the LRT1 heights of ERA5 is in the range of $\pm 200$ m at different latitudes. Therefore, a height difference of 250 m with respect to the tropopause is used as threshold for ERA5 data to identify stratospheric ice clouds in this study. One should keep in mind that gravity waves and deep convection are generally important factors influencing the height and variability of the tropopause (Sherwood et al., 2003; de la TORRE et al., 2004; Hoffmann and Spang, 2022).'

'As for the possible impacts of gravity waves and deep convection on the tropopause, Hoffmann and Spang (2022) found much more pronounced effects of gravity waves on the variability of tropopause heights and temperatures for ERA5 than ERA-Interim. However, convection-associated tropopause uplifts are not commonly represented, even in ERA5, due to the limited horizontal resolution of the reanalyses data sets. Since we used the same tropopause data set as Hoffmann and Spang (2022), tropopause uncertainties related to unresolved deep convection would exist in our study.'

7. Ln 157 CALIPSO averages their backscatter data over many profiles – this averaging is the effective along track resolution of the data. You should mention this.

We have revised in the manuscript. 'The vertical resolution of CALIPSO observations varies as a function of altitude. It is 60 m in the altitude range from 8.2 to 20.2 km. In the horizontal the profiles are averaged over 1 km along track distance between 8.2 km and 20.2 km of altitude.'

8. Ln 164 I don't see why you are including daytime aerosols. The S/N during the day drops significantly. If ice cloud detection is affected, aerosols will be worse since their backscatter cross section is smaller.

We agree. We excluded the daytime aerosol data from our study.

9. Ln 169 You should mention earlier that this study is not including PSCs.

We have revised the whole manuscript, presenting only data in $\pm 60°$ to avoid possible mixing of PSCs and SICs in the high latitudes.

10. Ln 180-190 I am surprised that you are using AIRS for deep convection. MODIS has the same channels (and others) and is more frequently cited for identifying cold cloud tops in the 8 and 10.6 µ channels. Furthermore, MODIS or VIIRS has higher spatial resolution.

Thank you, MODIS and VIIRS would be very helpful for deep convection analyses. But for now, we have only the whole time range of AIRS data at hand and the UTLS clouds retrieved from AIRS can well represent clouds from tropical storms and strong convective events at midlatitudes and high latitudes. MODIS or VIIRS data would be a good choice for our future work.

11. Ln 205 Gravity waves at 4µm are detected in the 30-45 km region (see Hoffmann and Alexander (2010), Fig. 3). These waves, propagating vertically, will have much lower amplitude at their tropopause source. For example, assume the wave is detected with a 0.5 K BT temperature (e.g. Hoffmann and Alexander Fig. 1) and using their the center of the weighting function, this wave is at 40km, then the amplitude of the wave at 18 km will be about 0.05K by energy conservation. There is also a spatial correction, as the wave move from the source it will decrease in amplitude. For example, let's say the wave is detected 1000km from the convective system and the convective system has a radius of 100km. Then the source wave will have an amplitude 3 times larger than the detected wave. The author needs to discuss these possible or indicate why they are unimportant corrections.

We agree it is important to better explain the measurement characteristics of the AIRS gravity wave observations and revised Sect. 2.3.2, accordingly. In particular, we added 'However, it is important to note that BT variances should not be confused with atmospheric temperature variances. The AIRS nadir observation geometry significantly reduces the sensitivity of the BT measurements compared to real atmospheric temperature fluctuations for short vertical wavelength waves. For the BT variances, the response to atmospheric temperature variances is near zero below 30 km of vertical wavelength and increases to about 50% at 65 km of vertical wavelength Hoffmann et al. (2014). With these measurement characteristics, AIRS is mostly sensitive to short horizontal and long vertical wavelengths waves, which are expected to propagate from the tropopause to the upper stratosphere within less than $1 - 2$ h and horizontal propagation distances less than a few hundred kilometers. The AIRS BT measurements should

be seen as a proxy of gravity wave activity.'

12. Ln 213 The difference between 7.1μ and 7.3μ bands as a method of estimating SO2 assumes you can neglect water vapor which will be important where the tropopause is below 8 km. Volcanic SO2 is more easily detected in the UV bands – OMI on Aura is making those measurements coincident with CALIPSO and AIRS and might be a better choice. I am still unsure why we should even use SO2 data since the aerosols come from CALIPSO and SO2 will not be evident in aerosols due to fires.

The aerosols from CALIPSO are used to explore their relationships with SICs. AIRS $SO_2$ measurements are an established method to measure volcanic plumes at day and night time (Hoffmann et al., 2014). Since we used CALIPSO nighttime data for SIC detection, we chose AIRS $SO_2$ data, as it provides nighttime information, in contrast to measurements using UV bands. We are interested in explosive volcanic eruptions with injections into the UTLS region, where AIRS $SO_2$ measurements are very sensitive. $SO_2$ data from AIRS are subsidiary used to identify volcanic injections that are related to the enhanced stratospheric aerosol load. To make this clear, we added the following sentences: 'In this work, an SI threshold of 10 K is applied to detect strong explosive volcanic eruptions with injections into the UTLS region.'

13. Ln 227 'over the tropical continents.'

Fixed.

14. Ln 228 Awkward wording... 'The weakest signal..'

We have revised it to 'the lowest frequency of '.

15. Ln 250 SIC associated with double tropopause are clearly an 'edge of the tropics' phenomenon – not surprising since that is where double tropopauses occur. I am not sure why these are important. They aren't contributing to dehydration of the stratosphere and double tropopauses are often associated with cloudy systems so I am not sure of their impact on the radiation budget...

We investigated SICs associated with double tropopauses, because they are closely related to the polar-ward isentropic transport and mixing of water vapor in the lowermost stratosphere. Double tropopause are associated with enhanced transport of water vapour from the tropics to the higher latitudes. Observations of thin ice clouds in the low stratosphere over the northern middle and high latitudes in August 1997 were traced back to tropical high-humidity air (Randel et al., 2007; Spang et al., 2015). Therefore, analyses of SICs associated with double tropopauses will help us understanding the air transport and water vapor variability in the UTLS. We added this discussion in the revised manuscript.

'Following the definition of the WMO, a second tropopause is identified if the average lapse rate at any level and at all higher levels within one kilometer exceeds 3° C/km above the first tropopause. The existence of a second tropopause indicates a less stable temperature structure in the UTLS region (Homeyer et al., 2014). Randel et al. (2007) discovered that the double tropopause indicates a region of enhanced transport from the tropics to higher latitudes. Thin ice clouds observed in the low stratosphere over the northern middle and high latitudes in August 1997 originated from tropical high-humidity air (Spang et al., 2015). Therefore, SICs detected in the vicinity of double tropopauses are probably related to quasi-isentropic transport of humid air from the tropics to the extratropics.'

16. Ln 275 Kim et al. emphasized the cooling rate, not just the low temperature was associated with cirrus. Obviously, you need to have a saturated environment for clouds to form, cooling the air creates saturation and ice crystals form. But once they form, they fall out so if the air is just cold and not cooling you don't see as many clouds. I think you should clarify this point.

Thank you. This point has been included in the revised manuscript. 'Low temperatures and cooling processes are more favorable for ice formation, and temperature normally has a negative relationship with cirrus cloud frequency (Eguchi and Shiotani, 2004; Kim et al., 2016).'

17. Ln 290, Ln 304-6 What is the explanation for positive correlations? SIC's show up where temperatures are warmer. Ln 306 appear to be guesses. You might consider that the positive correlations are associated with cooling air not cold air as suggested by Kim (see above).

We have revised the text and extended the discussion. The correlation coefficients between the four processes and the SIC frequencies are analyzed together in Sect. 3.7. The positive correlation coefficients between SICs and LRT1 temperature are found in the North American Monsoon and the Asian Monsoon, with high positive correlation coefficients between UTLS clouds, gravity waves, and stratospheric aerosols and SICs. The high SIC frequencies can be produced by ice and water vapor injection from UTLSs, air cooling induced by gravity waves, and more ice nuclei from stratospheric aerosols when the tropopause is warm.

'While the LRT1-T shows a general negative correlation, there are strong positive correlations over central America and the Caribbean Sea, Philippines and South Chinese Sea, and the Tibetan Plateau to the Caspian Sea.'

'The inherent correlations between all processes may help to explain the positive correlations between SICs and LRT1-T in the North American Monsoon and the Asian Monsoon regions. Even if the tropopause temperature is warm, UTLS clouds, gravity waves, and stratospheric aerosol could all contribute to the high occurrence frequency of SICs. For example, Fu et al. (2006) discovered that deep convection in the Asian Monsoon injected more ice and water vapor into the stratosphere with warmer tropopause temperatures. However, their strong correlation also makes it challenging to disentangle all processes' effects on the occurrence of SICs.'

'However, tropopause temperatures cannot explain some remarkable positive anomalies in SIC frequencies. For example, high SICs in November 2010 to January 2011, December 2011, March 2014, and April-May 2018 over the equator and high SIC anomalies in April-July 2011 at 5°N-20°N. We need to note that the cold temperature as well as the cooling of the atmosphere (Kim et al., 2016) are important for the variation of SICs. And the uplifting motions, gravity waves, the El Niño-Southern Oscillation (ENSO) and quasi-biennial oscillation (QBO) and potentially other effects would all impact the temperature and temperature variations (Abhik et al., 2019; Feng and Lin, 2019; Tegtmeier et al., 2020b) associated with SIC variability.'

18. Ln 335 Convection will push the tropopause upward. So what you are doing here is finding the deep convection using AIRS and then interpolating the ERA5 tropopause height onto the point, and if the cloud observations are above the ERA5 tropopause it is a SIC. How do you know you have correctly located the tropopause? In fact, you don't know if the cloud is above or below the tropopause and thus whether it is truly a SIC. You need a coincident temperature profile (perhaps GPS) to prove this. At least give the reader some idea of the uncertainty in these estimates for the tropopause in these cases. BTW, radar measurements over the US coincident with soundings show that convection drives the tropopause upward, collapses and leaves behind a residual cloud. You should quote some of these references to justify your assertions.

Yes, the uncertainties about the tropopause height related to deep convection are challenging to rule out in our data. We have added this discussion to the revised manuscript in Sect. 2.1 and 4.1, respectively.

'One should keep in mind that gravity waves and deep convection are generally important factors influencing the height and variability of the tropopause (Sherwood et al., 2003; de la TORRE et al., 2004; Hoffmann and Spang, 2022).'

'As for the possible impacts of gravity waves and deep convection on the tropopause, Hoffmann and Spang (2022) found much more pronounced effects of gravity waves on the variability of tropopause heights and temperatures for ERA5 than ERA-Interim. However, convection-associated tropopause uplifts are not commonly represented, even in ERA5, due to the limited horizontal resolution of the reanalyses data sets. Since we used the same tropopause data set as Hoffmann and Spang (2022), tropopause uncertainties related to unresolved deep convection would exist in our study. '

19. Ln 349-356 This discussion doesn't add anything to the paper.

This paragraph has been removed from the manuscript

20. Ln 364 'measurements are not significantly observed in tropics' ???

We have revised this sentence in the manuscript. 'Note that due to the wind filtering and visibility effects, gravity waves are not significantly observed in the tropics in AIRS (Hoffmann et al., 2013).'

21. Ln 358-384 Since SICs at high latitudes are correlated with cold temperature anomalies and cold temperature anomalies are correlated with gravity waves, I am not sure I understand how these terms are independent in Fig 5. In fact, the correlation pattern in Fig. 5 are almost identical. I suggest that

you present a map of correlations between temperature anomalies and gravity waves… This might give us more insight as to the causes. More specifically, cold temperature anomalies can be generated by mesoscale events as well as gravity waves. It is of interest to separate the two which I think might be lumped together in your analysis.

We agree that the cold temperature and gravity waves are strongly correlated. The individual plot of correlation coefficients between SICs and the tropopause temperature, gravity waves, UTLS clouds and stratospheric aerosols can show us the relationships of each process with SIC occurrence. As you pointed out, those individual correlation patterns present high correlations between all processes. However, we can't separate them currently. We have added the discussion of their high correlations in Sect. 3.7 in the revised manuscript.

'The occurrence of SICs has a general negative correlation with tropopause temperature, while SICs have positive correlations with UTLS clouds, gravity waves and stratospheric aerosols. The highest negative and positive correlations are mostly observed over the tropical continents and the western Pacific with correlation coefficients of $< -0.8$ between SICs and LRT1-T and $> 0.8$ between SICs and UTLS clouds, gravity waves, and stratospheric aerosols. High positive correlations are also found over the Asian Monsoon and the North American Monsoon regions between SICs and UTLS clouds, gravity waves, and aerosol. While the LRT1-T shows a general negative correlation, there are strong positive correlations over central America and the Caribbean Sea, Philippines and South Chinese Sea, and the Tibetan Plateau to the Caspian Sea. The highest correlation coefficients are as large as 0.8-1.0 in the North American Monsoon region, even for LRT1-T. In the Asian Monsoon region, negative correlations are detected over the Tibetan plateau, but positive correlations are seen over southern Asia and India between SICs and LRT1-T. High correlation coefficients imply the important role of tropopause temperature, UTLS clouds, gravity waves and stratospheric aerosols for the occurrence of SICs. However, overlapping high correlation coefficients indicate also strong connections between the tropopause temperature, UTLS clouds, gravity waves, and stratospheric aerosols themselves.

To further investigate the source of SICs, the highest and second-highest correlation coefficients between SICs and all processes for each grid box are shown in Fig. 11. Over the tropical continents, the highest correlation coefficients of SICs relate to tropopause temperature. The highest correlation coefficients are found between UTLS clouds and SICs in the monsoon domains in the latitude range between 15° and 30°, e.g., the North American Monsoon, the Asian Monsoon, the South African Monsoon regions and the La Plata basin. In the central United States, tropopause temperature and UTLS clouds have the highest correlations with SICs. Over Patagonia and the Drake Passage, tropopause temperature and gravtiy waves have the highest correlation with the occurrence of SICs. In the latitude range between 45° and 60°, the strongest correlations are found between SICs and tropopause temperature and gravity waves. However, the second-highest correlation coefficients of SICs are related to stratospheric aerosols, UTLS clouds, and gravity waves over the tropical continents, the North American Monsoon and the Asian Monsoon regions. The rather similar correlation coefficients of SICs with all processes indicate high correlations between all processes themselves.

For all processes, increased tropopause-penetrating convection may result in a cooler tropopause across the tropics (Gettelman et al., 2002). Gravity waves and wave breaking will locally cause a colder temperature in the atmosphere and air cooling (Dinh et al., 2016). High correlations were found between deep convection and gravity waves (Hoffmann et al., 2013), and vertical motion of air will transport aerosols into the stratosphere (Bourassa et al., 2012). The inherent correlations between all processes may help to explain the positive correlations between SICs and LRT1-T in the North American Monsoon and the Asian Monsoon regions. Even if the tropopause temperature is warm, UTLS clouds, gravity waves, and stratospheric aerosol could all contribute to the high occurrence frequency of SICs. For example, Fu et al. (2006) discovered that deep convection in the Asian Monsoon injected more ice and water vapor into the stratosphere with warmer tropopause temperatures. However, their strong correlation also makes it challenging to disentangle all processes' effects on the occurrence of SICs.'

22. Ln 385 I am not sure why you are even using the AIRS $SO_2$ since CALIPSO measurement of aerosols is more of a direct measure of the influence of aerosols on cloud formation. Furthermore, aerosols are also due to fires (see the anomaly at end of the period shown in Fig. 10 which is the Australian fires.) – see major comments.

Please see answers to general comment no. 4 and specific comment no. 12. CALIPSO aerosol contains aerosol of various origins. To get an idea, which aerosol coincides with SICs we used AIRS SO$_2$ data to mark volcanic eruptions. This shows that mainly volcanic aerosol coincides with SIC anomalies, but also wild fires, such as in 2017 over North America. However, the Australian bush fires 2009 and 2019/2020 do not coincide with SIC anomalies.

23. Ln 427 Spearman

Fixed.

24. Ln 460 Table not Tab.

Fixed.

25. Ln 467 where did 'Atmospheric Turbulence' estimation come from?

We have revised this sentence in the manuscript.

26. Ln 481, 485 Fig. 6a

Revised.

27. Ln 475 to 485. This is a very interesting discussion but still does not incorporate the point that deep convection can push up the tropopause so that SIC observations that are apparently above the tropopause are actually in the troposphere. Unless a GNSS RO measurement is made exactly at the right spot, the ERA5 reanalysis will place the tropopause at the wrong altitude.

Please see answers to specific comment no. 18. The discussion of tropopause uncertainties related to deep convection has been added to the revised manuscript in Sect. 2.1 and 4.1. The tropopause height uncertainties aren't yet ruled out in our tropopause data.

'One should keep in mind that gravity waves and deep convection are generally important factors influencing the height and variability of the tropopause (Sherwood et al., 2003; de la TORRE et al., 2004; Hoffmann and Spang, 2022).'

'As for the possible impacts of gravity waves and deep convection on the tropopause, Hoffmann and Spang (2022) found much more pronounced effects of gravity waves on the variability of tropopause heights and temperatures for ERA5 than ERA-Interim. However, convection-associated tropopause uplifts are not commonly represented, even in ERA5, due to the limited horizontal resolution of the reanalyses data sets. Since we used the same tropopause data set as Hoffmann and Spang (2022), tropopause uncertainties related to unresolved deep convection would exist in our study. '

28. Ln 494 It seems to me that the event frequency will over emphasize the SIC occurrence above convection. If you're trying to connect these observations with stratospheric hydration (which is why I am interested in this) then occurrence frequency is the appropriate measure. If you are interested in the morphology of these events, then event frequency is appropriate. It might help to discuss some of these issues at the beginning of this section to motivate the reader.

The event frequency is defined as the ratio of number of days in which UTLS clouds or SICs ($\geq 1$ detection) occurs to the total number of days in a given time period over a given region, which is used to eliminate the morphological effects of UTLS clouds in this study. The clarifications have been extended in the revised manuscript.

'Next to the occurrence frequencies, the event frequency is defined in this work as the ratio of number of days in which UTLS clouds or SICs ($\geq 1$ detection) occur to the total number of days in a given time period over a given region. The event frequency helps overcome some of the limitations related to cloud geometries for UTLS clouds and SICs.'

29. Ln 511 OMPS-LP makes aerosol limb measurements and is active for most of the period you studied.

Thank you. The suggested OMPS-LP would be very helpful for aerosol studies. We will consider this high vertical resolution data in our future work.

**References**

[revised manuscript text omitted]

---

## Author Comment (AC2)

**Reviewer #1**

**General Comments**

This article builds upon Zou et al. (2020) and Zou et al. (2021), by mostly the same authors, which I also reviewed. In their 2021 article, the authors contrasted the occurrence of stratospheric cirrus clouds seen by CALIPSO above the United States with deep convection and gravity waves activity as seen by AIRS. The article under review applies more or less the same methodology to the global scale (also updating their tropopause retrievals to ERA5 from ERA-Interim).

The article under review is quite long, includes many figures and contains interesting pieces of information. Its structure and writing are easy to follow. It is however, in my opinion, quite sprawling, and fires in too many directions. The lack of focus means this reader was often confused. Some choices of figures are not optimal, and I have one major methodological concern. Below I attempt to explain where my confusion came from and suggest ways for improvement.

**Answer:** Thank you very much for your time and effort in reviewing the present and our previous manuscripts. We considered all the comments and suggestions and revised the entire manuscript.

**General Comments ∗ 1**

I will start with what I think is a methodological problem. On line 489, you state that "clouds are labeled as deep convection if the cloud top brightness temperature is close to the tropopause temperature with an offset of 7K from the tropopause temperature". If I understand this, and the explanations of Sect. 2.3, correctly, "deep convection" then means "AIRS has observed a cloud with a top temperature close to (ie as high as) the tropopause temperature". In other words, "deep convection" is not derived from a dynamical measurement, but implied by the detection of a high-altitude cloud (taking into account the detection sensitivity of AIRS). Thus your comparison of stratospheric ice clouds (from CALIOP) with "deep convection" (from AIRS) is really a comparison of stratospheric clouds (seen by CALIOP) with high tropospheric clouds (seen by AIRS), i.e. a comparison of cloud detection sensitivities of both instruments. Differences between both retrievals will be mostly attributable to instrument strengths and weaknesses considering various cloud geometries, and not to processes that might lead to formation of stratospheric cloud (like convection). I would appreciate if you could address this point and either take it into account somehow it in your comparison, or justify why you think your comparison goes beyond a comparison of instrument sensitivities. Moreover, this issue creates a problem for the definition of deep convection. AIRS detecting a high tropospheric cloud in the tropics can indeed imply "deep convection" is occurring (as deep convection is probably the mechanism that brought the observed cloud near the tropopause), but in the midlatitudes and the polar regions convection is not required to create clouds near the tropopause. Calling AIRS high tropospheric cloud detections "deep convection" outside the tropics feels wrong to me. Considering this, you might want to change the name of "deep convection" in the paper to something else ("high tropospheric clouds" ?).

**Answer:** By using the brightness temperature threshold of $+7\,\mathrm{K}$ above the $\mathrm{T}_{TP}$, we identified high altitude clouds from AIRS that have a significant optical thickness with tops in the upper troposphere and lower stratosphere (UTLS clouds). In the tropics, most tropopause reaching clouds with large optical thickness could be related to a deep convection origin. However, in the midlatitudes, high altitude clouds, which are thick enough to be detected by AIRS, can be also related to frontal systems (warm front uplifting), jet stream, mountain wave and contrails. We have changed the 'deep convection' to 'UTLS clouds' and clarified its definition in the manuscript.

The comparison of SICs measured by CALIPSO and UTLS clouds measured by AIRS can be affected by their detection sensitivities. However, the thick UTLS clouds detected in AIRS can well represent fresh deep convection in the tropics. Therefore, UTLS clouds are used as a proxy for deep convection and other high altitude cloud sources in this study. The term 'event frequency' is employed in our study to eliminate somehow the impacts of clouds geometries. The event frequency is the ratio of number of days in which convective system clouds ($\geq 1$ detection) occur to the total number of days in a given time period over a given region.

'In this study, temperature differences between AIRS brightness temperatures ($BT_{AIRS}$) and tropopause temperatures ($T_{TP}$) from ERA5 are employed to detect high altitude clouds in the tropics and at midlatitudes. A threshold of $+7\,K$ above $T_{TP}$ was chosen to identify possible high altitude clouds with tops in the upper troposphere and lower stratosphere, also referred to as UTLS clouds (Zou et al., 2021). In the tropics, most tropopause-reaching clouds with large optical thickness could be related to a deep convection origin (Gettelman et al., 2002; Tzella and Legras, 2011). At midlatitudes, high altitude clouds could also be related to frontal systems (warm front uplifting), mesoscale convective systems and mesoscale convective complexes, jet stream, mountain wave and contrails (Field and Wood, 2007; Trier and Sharman, 2016; Trier et al., 2020). UTLS clouds are considered here as a proxy for deep convection in the tropics and other high altitude ice cloud sources at midlatitudes.

Next to the occurrence frequencies, the event frequency is defined in this work as the ratio of number of days in which UTLS clouds or SICs ($\geq 1$ detection) occur to the total number of days in a given time period over a given region. The event frequency helps overcome some of the limitations related to cloud geometries for UTLS clouds and SICs.'

**General Comments ∗ 2**

The second issue I have with the paper is that the subject under study is often unclear. The title mentions stratospheric ice clouds. But are we talking about clouds injected by convective overshoots in the Tropics, which are a potential pathway for stratospheric hydration? Are we talking about clouds generated through gravity wave-induced cooling over the polar regions, a subtype of polar stratospheric clouds which can also include nitric or sulfuric acid? Those objects are not the same, they do not occur in the same regions and in the same thermodynamic conditions, they do not follow from the same physical processes, they do not affect the atmosphere in the same way, they do not lead to the same scientific questions. In my view, it is not possible to study those clouds as if they were the same thing, as it is attempted here. Due to this mixing up of very different objects, the article sometimes discusses issues in regions that are irrelevant.

Related to this second point, I have strong concerns about how results at polar latitudes are presented together with results at lower latitudes – for instance maps of deep convection events that extend to polar regions. Given the results, I am quite convinced polar stratospheric clouds represent a non-negligible part of the dataset (maybe the whole thing) at high latitudes, even though some of them are filtered out. Presenting results that are partially representative of PSCs in maps that mainly target stratospheric ice clouds linked to deep convection I find particularly problematic. The paper also does not make it clear enough if it wants to consider PSCs as part of the dataset under study, or if it wants to keep them out of the dataset under study. Below I suggest to exclude PSCs not only from the dataset but also from the results presented. This will help the paper clarify its object of study. To fix this second issue, how I see it, the paper could 1) strive harder to eliminate PSC for the input dataset and from the presented results, 2) better explain what is meant by "deep convection" here, as it does not follow the usual convention (note that my first comment above somewhat clarifies that confusion), and 3) either simplify the analysis by removing some figures or split the article in two (maybe move the regional analyses in its own paper).

**Answer:** Thank you for your helpful comment. We focus only the stratospheric ice clouds in this study. The objective is to investigate the distribution and variation of SICs, and to explore potential

relationships between the occurrence of SICs and atmospheric processes, i.e., atmospheric temperature, isentropic transport, stratospheric aerosols, convective system clouds, and gravity waves.

(1) We have revised all figures and results in the new manuscript and limited our analyses to the tropics and midlatitudes (±60°) to avoid interferences with unfiltered PSCs.

(2) We have changed the 'deep convection' to 'UTLS clouds' and clarified its definition in the manuscript.

(3) We have reorganized the manuscript, removed the regional analyses, and added some conclusive discussion in Sect. 3.7 in the revised manuscript.

' A filter criterion for polar stratospheric clouds (PSC) (Sassen et al., 2008), i.e., data are excluded if CTHs are higher than 12.0 km in areas with local winter latitude ≥ 60° N and 60° S, is utilized here to avoid possible miscounting of PSC. However, this filter criterion may not catch all low altitude PSCs. Therefore, we limited our analyses to the latitude range of ±60°.'

'In this study, temperature differences between AIRS brightness temperatures ($BT_{AIRS}$) and tropopause temperatures ($T_{TP}$) from ERA5 are employed to detect high altitude clouds in the tropics and at midlatitudes. A threshold of $+7$ K above $T_{TP}$ was chosen to identify possible high altitude clouds with tops in the upper troposphere and lower stratosphere, also referred to as UTLS clouds (Zou et al., 2021). In the tropics, most tropopause-reaching clouds with large optical thickness could be related to a deep convection origin (Gettelman et al., 2002; Tzella and Legras, 2011). At midlatitudes, high altitude clouds could also be related to frontal systems (warm front uplifting), mesoscale convective systems and mesoscale convective complexes, jet stream, mountain wave and contrails (Field and Wood, 2007; Trier and Sharman, 2016; Trier et al., 2020). UTLS clouds are considered here as a proxy for deep convection in the tropics and other high altitude ice cloud sources at midlatitudes. '

**General Comments ∗ 3**

The third issue I have with the paper is that it investigates the role of many processes (deep convection, gravity waves, stratospheric aerosols loading from eruptions  biomass burning) in the formation of stratospheric ice clouds, but never attempts to summarize its findings in an integrated view, that would for instance rank the importance of each of these processes spatially or temporally. No attempt is made to provide a theoretical framework that would justify why stratospheric clouds are more frequent when a given process is more frequent in a given region or period, and why they are more frequent when another process is more frequent in another region or period. The many figures often plot the evolution of a property of stratospheric cloud against the evolution of another value representative of a process. Helped by the text, we see when/where both values are more correlated, and when/where they are less correlated, but as readers, we are just left with correlation coefficients, with no improved understanding of what is going on. This problem is somehow compounded by the first issue – investigating so many subjects makes it more obvious that no attempt is made to bring them all into a cohesive whole. Making the article(s) more focused would make it easier to integrate findings into a larger context.

**Answer:** Thank you for this comment. We have reorganized and revised the entire manuscript. This study aims to understand the relationships between SIC occurrences and related processes, e.g., tropopause temperature, double tropopauses, UTLS clouds and stratospheric aerosols.

We investigated the individual and combined relationships between different processes and SIC occurrence. 'Correlation coefficient and long-term anomaly analyses of SICs and all the other processes indicate that the occurrence and variability of SICs are mainly associated with the tropopause temperature in the tropics. UTLS clouds have the highest correlations with SICs in the monsoon regions and the central United States. Tropopause temperature and gravity waves are mostly related to SICs at midlatitudes, especially over Patagonia and the Drake Passage. However, besides the highest correlation coefficients, the cold tropopause temperature, the occurrence of double tropopauses, high stratospheric aerosol loading, frequent UTLS clouds and gravity waves all have high correlations with the SICs. The occurrence and variability of SICs demonstrate a strong dependence on various processes, both locally

and temporally.

The overlapping and similar correlation coefficients between SICs and all processes indicate strong associations between all processes themselves. Due to their high inherent correlations, it is challenging to disentangle and evaluate their contributions to the occurrence of SICs on a global scale. However, the correlation coefficient analyses between SICs and all processes and high associations between all processes observed in this study help us better understand the sources of SICs on a global scale.'

**Specific Comments:**

1. l. 13-20: The past time makes it unclear who did the things explained. "Relations... were analyzed" – analyzed by who?

Fixed.

2. l. 47: "For example, 7 % of observations..." in your 2021 paper, the number given was 2.5 %. Which is correct?

Both numbers are correct. I should have been more specific. '7 % of observed clouds (Noël and Haeffelin, 2007, Table. 1) and 2.5 % of observations (Noël and Haeffelin, 2007, Fig. 3)'. This sentence has been removed from the manuscript due to the revision of this paragraph.

3. l. 148: "in previous studies..." this sentence suggests that the cited articles used ERA-Interim to derive the tropopause altitude. Please mention it explicitly. Also: it's unclear to me why a 2x resolution improvement means the threshold for tropopause can also be cut in half. Other considerations than vertical resolution influence the accuracy of the retrieved tropopause, and the distance you wish to impose on that tropopause to make sure one is in the stratosphere. Would ERA6 improve the vertical resolution 10x, you would still need to consider a larger threshold to account for other sources of uncertainty.

Sorry for the confusing text, the 2x resolution improvement coincides with the 250 m. The 250 m is set as the tropopause threshold because the tropopause height differences between ERA5 and radiosonde and GPS are about 200 m. We have revised the manuscript.

'Tegtmeier et al. (2020a) found that LRT1 height differences between ERA5 and Global Navigation Satellite System-Radio Occultation observations are less than 200 m in the tropics. Based on US High Vertical Resolution Radiosonde Data (HVRRD) data and coarser-resolution Global Positioning System (GPS) data, Hoffmann and Spang (2022) also showed that the uncertainty of the LRT1 heights of ERA5 is in the range of ±200 m at different latitudes. Therefore, a height difference of 250 m with respect to the tropopause is used as threshold for ERA5 data to identify stratospheric ice clouds in this study. '

4. Sect. 2.2: in this section, you explain how from the CALIPSO VFM product you use cloud data, but also aerosol data. I must admit I was confused at that point since I had missed that one of the objectives of the paper was to contrast the presence of stratospheric ice clouds with the occurrence of stratospheric aerosols. As far as I can tell, the paper's objectives are only explained in the sentence l. 122-124. It might be good to expand a bit this sentence to make sure other readers will not miss it.

Done. Aerosols are effective ice nuclei for cirrus cloud formation and variation. Therefore, stratospheric aerosols are investigated in this study to explore their potential impacts on the occurrence of SICs.

'The objectives of this study are to 1) examine the distribution and long-term variation of stratospheric ice clouds and 2) investigate potential effects of atmospheric temperature, stratospheric aerosols, UTLS clouds, and gravity waves on the occurrence and distribution of SICs on a global scale.'

'As aerosol particles provide cloud condensation nuclei and ice nuclei, the occurrence of SICs is expected to correlate with aerosols (Lohmann and Feichter, 2005).'

5. l. 163-165: I don't follow the reasoning that justifies why daytime and nighttime data are used for aerosols but only nighttime data for ice clouds. Why should "aerosols are long-lived" justify using both daytime and nighttime data? Why is increased nighttime SNR (=improved detection abilities) a good reason to use only nighttime data for ice clouds, but not for aerosols? Will not this difference in datasets influence somehow the comparisons between stratospheric clouds and stratospheric aerosols? Please clarify your reasons why keeping only the nighttime data for clouds and using both daytime and nighttime data for aerosols.

To ensure the consistency of data, we have removed the daytime aerosol data from the manuscript.

6. l. 169: PSCs often reach latitudes lower than 60°. In the north hemisphere, they were observed as far down as the Mediterranean sea : https://doi.org/10.5194/acp-7-5275-2007 I'm afraid your criteria to exclude PSCs will still lead to a large presence of PSCs in your stratospheric cloud dataset, and the presented results suggest this to be true. I suggest that PSC *are* stratospheric clouds, and the distinction you're trying to make here between PSC and other stratospheric ice clouds is not really possible – above a given latitude, all stratospheric clouds are PSC, by definition. But not all PSCs are ice. It is unclear to me if the paper wants to include ice PSCs within the boundaries of the dataset under study, or not. Limiting the geographic scale of the study, for instance to only show latitudes below 60°, would put a limit on the importance of PSCs in the dataset considered, and in the results presented.

Thank you. The PSCs are not supposed to be investigated in this study. Even though the filter criteria were used to filter polar stratospheric clouds, some low-altitude PSCs were still obtained in the data. In the revised manuscript, we have limited the analyses to the tropics and midlatitudes (±60°).

7. l. 171: "SICs detected at high latitudes will not be discussed in detail in this work". This is good, but all your maps still show latitudes and results above 60°. For example, the same map will mix stratospheric ice clouds along with polar stratospheric ice clouds. However, at high latitudes, only parts of the observed PSCs are shown, given the filtering described on l. 169. Thus what global maps show at high latitudes is neither representative of convection-based ice clouds, nor of polar stratospheric clouds, and I'm concerned these figures could at a glance be misinterpreted by a too-quick reader. I would appreciate if high latitudes, where the dataset under study is probably dominated by PSCs but omits an undefined amount of them, where hidden from the global maps.

We have limited the analyses to the tropics and midlatitudes (±60°). All data and figures are revised in the new manuscript.

8. Sect. 2.3: Many parts of this section are very similar to section 2.3 of Zou et al. 2021 (a section with more or less the same name). I've found at least one sentence that is exactly the same. Please revise this part to see if you could perhaps just reference the previous paper.

We have revised this section. Please find revisions in the new manuscript.

9. 181: could you please be more specific when you reference the 7 articles here? ie cite each paper separately when it is most relevant.

Done. We have separated them into three subsections with corresponding references.

'Aumann et al. (2006) and Aumann et al. (2011) retrieved deep convective clouds from AIRS at 8.1 $\mu$m (the 1231 cm$^{-1}$ atmospheric window channel) in the tropics. The term "deep convective clouds" in their studies refers to clouds tops of thunderstorms in non-polar regions with a brightness temperature (BT) of less than 210 K. When the top of anvil of thunderstorms has a brightness temperature of less than 210 K, the deep convective clouds are considered to reach the tropopause region in the tropics (Aumann et al., 2006). However, the threshold of 210 K is too low for midlatitude convective events (Hoffmann and Alexander, 2010), and a constant brightness temperature threshold for convective event detection may produce ambiguous results at different latitudes and seasons (Hoffmann et al., 2013).'

'In this study, mean variances of detrended brightness temperatures in the 4.3 $\mu$m carbon dioxide waveband are used to identify stratospheric gravity waves from AIRS observations (Hoffmann and Alexander, 2010; Hoffmann et al., 2013). Measurements of 42 AIRS channels from 2322.6 to 2345.9 cm$^{-1}$ and 2352.5 to 2366.9 cm$^{-1}$ are averaged to reduce noise and improve the detection sensitivity of the gravity wave observations. Even though the AIRS observations have the highest sensitivity at an altitude range of $30-40$ km (Hoffmann and Alexander, 2010; Hoffmann et al., 2013, 2018), the averaged BT variance can provide gravity wave information for the lower stratosphere as gravity waves typically propagate upward from the tropospheric sources into the stratosphere.'

'As brightness temperature differences are an effective method to detect volcanic $SO_2$ from AIRS observations (Hoffmann et al., 2014b, 2016), spectral features of $SO_2$ at 1407.2 $cm^{-1}$ and 1371.5 $cm^{-1}$ are used to calculate the $SO_2$ Index (SI)'

10. l. 221: "lower stratospheric ice clouds": What does "lower" mean here? I don't think low stratosphere and high stratosphere were defined so far in the paper.

Revised. 'Global stratospheric ice clouds'

11. l. 222: "Ice clouds with cloud top heights at least 250m above the first tropopause were defined as SICs" I think this has already been explained.

We have removed this sentence from the manuscript.

12. l. 248: "The occurrence of double tropopauses in general greatly impacts the SICs' occurrences associated with double tropopauses. " I'm not sure I understand this sentence. As I read it, I think it means that when there is no double tropopause, there are no SIC associated with a double tropopause? Please clarify.

We have revised this subsection. This sentence is removed from the manuscript.

13. Figure 1. These figures show SICs are quite frequent over the Antarctic Peninsula in all seasons except DJF (Antarctic summertime). This supports the idea that the SIC dataset includes a non-negligible part of PSCs.

We have limited the analyses to the tropics and midlatitudes ($\pm 60°$). All data and figures are revised in the new manuscript.

14. l. 258: thermal tropopauses are notably hard to retrieve over the polar regions, where the temperature gradient gets mostly flat in the stratosphere. It is not clear to me that retrievals of multiple tropopauses in those regions are neither reliable nor meaningful. What is the meaning of mutiple tropopauses when the temperature profile is flat?

We have excluded high latitude data in the revised manuscript.

15. Figure 2: In this figure, I doubt the relevance of multiple tropopauses that appear above 60°N and above 60°S. See previous comment.

We have excluded high latitude data in the manuscript.

16. Figure 3: This figure is very pretty, but it does not support the discussion very well. The text discusses the differences between years, and the average yearly evolution of the SIC frequencies. This discussion would be better supported by showing the average yearly evolution of SIC occurrence (ie the same Hovmoller plot as Figure 3 but for months averaged over 2007-2019), and maybe in addition a plot showing the monthly anomalies of SICs occurrence averaged over the ±20° region (I'm not sure the discussion discusses the small variations that occur outside of this latitude range).

We have revised the corresponding text for this figure. We want to present the seasonal cycles and inter-annual variability of SICs over different latitude bands using a Hovmöller diagram. We can easily see the variation of SICs over latitude ranges and the time series from a Hovmöller figure.

'To investigate spatial and temporal variations of SICs, monthly averaged occurrence frequencies of SICs in 5° latitude bands from 2007 to 2019 are shown in Fig.2. Seasonal cycles and inter-annual variability of SICs are observed in the tropics and at midlatitudes. SICs in the tropics follow the Intertropical Convergence Zone (ITCZ) over time, i.e., high SIC frequencies in the latitude range of 20° S-20° N move from south to north from boreal winter to summer and north to south from boreal summer to winter. The correlation with the ITCZ suggests that there is a strong correlation with deep convection. Most SICs are observed between 15° S-5° N, which show higher SIC occurrence frequencies ($> 0.24$) and longer occurrence times (November to March of the following year). The SIC frequencies are stronger in the SH tropics, whereas SICs extend to higher latitudes in the Northern Hemisphere. Some SICs are identified at 25° N-30° N from June to August, which are absent in the Southern Hemisphere, which would relate to the uplift of the Tibetan Plateau and the Asian Monsoon region.

At midlatitudes, the frequencies of SICs at midlatitudes are at least twice as low as in the tropics. However, we can still notice an inter-annual variation of SICs at midlatitudes, where SICs are more often observed in winters/early springs. It suggests other sources for the occurrence of SICs at midlatitudes besides deep convection. Therefore, we investigate the correlation of different processes with respect to SIC occurrences in the following sections, including tropopause temperature, double tropopauses, UTLS clouds, gravity waves, and stratospheric aerosol, which are expected to have an impact on cloud formation. '

17. Section 2.3: I find this section quite confusing. It does not include the easiest way to look for a

correlation: a scatterplot. Why not plot first the frequencies of SIC in 5x10° cells against the average tropopause temperature in the same cells? This would let you first conclude on the existence of a correlation between both quantities. Then figure 5 would let you identify regions where the correlation is positive and where it is negative, and finally figure 6 would let you identify possible variations of these correlation signs in time.

The manuscript has been restructured. The seasonal maps of LRT1 temperature and SIC occurrence frequencies are presented in Sect. 3.3 to indicate their possible correlations. A scatter plot could show a correlation, but it may miss the seasonal dependency in the tropics as explained in the text. The correlation coefficients between processes and SIC occurrence frequencies and long-term spatial variability of SICs and all processes are presented in Sect. 3.7 to investigate the relationships between all processes and SICs.

18. l. 289: I understand that a positive correlation means that SIC are more frequent when the tropopause is warm, and less frequent when the tropopause is cold. If that is your understanding too, could you propose some kind of explanation or process responsible for positive correlation? This result is clearly at odds with the findings of the articles you cite, and cannot go by unadressed.

We have revised the text and extended the discussion. The correlation coefficients between the four processes and the SIC frequencies are analyzed together in Sect. 3.7. The positive correlation coefficients between SICs and LRT1 temperature are found in the North American Monsoon and the Asian Monsoon, with high positive correlation coefficients between convective system clouds, gravity waves, and stratospheric aerosols and SICs. The high SIC frequencies can be produced by ice and water vapor injection from convective systems, air cooling induced by gravity waves, and more ice nuclei from stratospheric aerosols when the tropopause is warm.

'While the LRT1-T shows a general negative correlation, there are strong positive correlations over central America and the Caribbean Sea, Philippines and South Chinese Sea, and the Tibetan Plateau to the Caspian Sea.'

'The inherent correlations between all processes may help to explain the positive correlations between SICs and LRT1-T in the North American Monsoon and the Asian Monsoon regions. Even if the tropopause temperature is warm, UTLS clouds, gravity waves, and stratospheric aerosol could all contribute to the high occurrence frequency of SICs. For example, Fu et al. (2006) discovered that deep convection in the Asian Monsoon injected more ice and water vapor into the stratosphere with warmer tropopause temperatures. However, their strong correlation also makes it challenging to disentangle all processes' effects on the occurrence of SICs.'

'However, tropopause temperatures cannot explain some remarkable positive anomalies in SIC frequencies. For example, high SICs in November 2010 to January 2011, December 2011, March 2014, and April-May 2018 over the equator and high SIC anomalies in April-July 2011 at 5°N-20°N. We need to note that the cold temperature as well as the cooling of the atmosphere (Kim et al., 2016) are important for the variation of SICs. And the uplifting motions, gravity waves, the El Niño-Southern Oscillation (ENSO) and quasi-biennial oscillation (QBO) and potentially other effects would all impact the temperature and temperature variations (Abhik et al., 2019; Feng and Lin, 2019; Tegtmeier et al., 2020b) associated with SIC variability. '

19. figure 5: When first reading section 3.4, only figure 5a is discussed. The other three figures are discussed further down. This is unusual and should be adressed somehow in the text or the legend.

We have reorganized the manuscript. The figure was moved to a new subsection, and all subplots were discussed together. Please find revisions in Sect. 3.7.

20. l. 296-300: providing pseudocode is not required for such a simple operation. The explanation lines 294-295 is enough.

The pseudocode has been removed.

21. figure 6: like with figure 3, I'm not sure the plot supports the discussion in an optimal way. The anomalies in SIC frequency shown in Figure 6A are very weak almost everywhere except in the ±20° band – most of the plot is not useful. In my view it would be much more readable to provide simpler line plots that describe the average anomaly in different latitude bands – for example one line for

$\pm 20°$, one for $> 20°$N and one for $< 20°$S. The same quantity of information would be offered, and the correlation/anticorrelation with the temperature anomalies would be much easier to see.

We have adjusted the color palette of the Hovmöller diagrams and reorganized them in Fig. 12. The Hovmöller figures are still used in our manuscript because we think they can give more detailed information about the latitude and time range of the anomalies, which helps us to better understand the related processes. Discussions on Hovmöller figures have been extended in the revised manuscript.

'To explain the tempo-spatial variation of SICs, monthly SIC frequencies and all processes at different latitude bands ($5°$ for each band) from 2007 to 2019 are presented in Fig. 12. The monthly anomalies for each band were computed as the difference between the monthly zonal mean values and the inter-annual mean of the monthly zonal mean values, which excludes seasonal cycles of parameters. The regionally averaged monthly anomalies of SIC frequencies and all processes with seasonal cycles over the tropics ($20°$S-$20°$N), northern midlatitudes ($40°$N-$60°$N) and southern midlatitude ($40°$S-$60°$S) can be found in Appendix.D.

For global-scale anomalies excluding the effect of seasonal cycles, significant anomalies in SIC frequency can be observed in the tropics. Anomalies of SIC frequencies at $\pm 20°$ are generally demonstrating contrary features to the LRT1 temperature. For instance, negative anomalies of SICs in February 2007 to July 2007, November 2009 to January 2010, October 2013 to June 2014 (excluding March 2014), and October 2015 to January 2016, January-August 2017, November-December 2019 are compatible with positive LRT1 temperature anomalies. And positive SIC anomalies in January-June 2008, January 2013, June-July 2015, June-December 2016, October 2018 to February 2019 are co-located with negative LRT1 temperature anomalies. During those periods, tropopause temperature variations are important for the anomalous variability of SICs in the tropics.

However, tropopause temperatures cannot explain some remarkable positive anomalies in SIC frequencies. For example, high SICs in November 2010 to January 2011, December 2011, March 2014, and April-May 2018 over the equator and high SIC anomalies in April-July 2011 at $5°$N-$20°$N. We need to note that the cold temperature as well as the cooling of the atmosphere (Kim et al., 2016) are important for the variation of SICs. And the uplifting motions, gravity waves, the El Niño-Southern Oscillation (ENSO) and quasi-biennial oscillation (QBO) and potentially other effects would all impact the temperature and temperature variations (Abhik et al., 2019; Feng and Lin, 2019; Tegtmeier et al., 2020b) associated with SIC variability.

Stratospheric aerosols, UTLS clouds and gravity waves are further analyzed to understand those anomalous SICs. Enhanced stratospheric aerosols due to volcanic eruptions coincide with the high SIC frequencies at $25°$S-$10°$N in November 2010 to January 2011 (Merapi volcano), $5°$N-$20°$N in April-July 2011 (Nabro volcano), $15°$S-$10°$N in March 2014 (Mt. Kelud volcano), $15°$S-$20°$N in April-May 2018 (Ambae volcano) (Global Volcanism Program, 2013; Hoffmann, 2021). In the extra-tropics, the most pronounced positive anomalies in SIC frequency correlate with the ash rich volcanic eruptions of Kasatochi (August 2008, $52°$N), Puyehue-Cordón Caulle (June 2011, $41°$S), Calbuco in April-May 2015 ($41°$S), and Raikoke (June 2019, $48°$N) (compare with AIRS ash and $SO_2$ index Hoffmann, 2021). High SIC frequencies around $40°$N in January-March 2011 and from December 2012 to January 2013 are co-occurring with high anomalies of UTLS clouds and gravity waves. The tempo-spatial analyses of LRT1 temperature, UTLS clouds, gravity waves and stratospheric aerosols provide explicit awareness of processes on the occurrence and variability of SICs at different latitude bands and time ranges.'

22. l. 307-308: "tropopause temperatures are negatively correlated with the occurrence of SIC... especially over tropical continents" you already concluded that from figure 5. Compared to figure 5, figure 6 adds the time periods and latitude bands where the correlation was positive and where it was negative. This is what should be discussed when focusing on this figure.

We have reorganized the Hovmöller diagrams in Fig. 12. The Hovmöller figures are discussed in more detail in the revised manuscript. Please see the answer to Point 21.

23. l. 314: In my understanding, deep convection is triggered by strong sunlight over water vapor-saturated areas, and occurs primarily in the tropics, especially in the warm pool and over the ITCZ – see for instance https://doi.org/10.1016/j.atmosres.2020.105244 that provides an example of such a definition. "Deep convection" then means convection that is triggered near the surface and generates

vertical motion all the way up to the tropopause (hence the "deep"). In your plots (figure 7), the warm pool appears as a minor hotspot of deep convection, and the ITCZ is not really visible. Moreover, your plots suggest that deep convection is very frequent over, for example, the northern Atlantic (30°N-70°N) – all during the cold, sun-deprived wintertime (DJF). I find all this very puzzling, and would appreciate if you could clarify the meaning of "deep convection" that you support in this article. Could you perhaps cite articles that support this definition? (note that my first major comment somewhat fixes that confusion, by clarifying that "deep convection" in the text really means "a cloud observed above the tropopause by AIRS")

We have changed the 'deep convection' to 'UTLS clouds' in the new manuscript. UTLS clouds retrieved from AIRS are used as a proxy for deep convection and other high altitude cloud sources in this study. UTLS clouds include clouds from tropical storms and strong convective events from numerous sources, such as storm systems and fronts, mesoscale convective systems, and mesoscale convective complexes in midlatitudes and high latitudes. Please see the answer to General Comment 1.

24. section 3.4: Figure 9 clearly shows that the gravity waves considered in the present article (and derived from AIRS) are mostly related to the presence of the polar vortex – polar gravity waves above 60N in DJF and above 60S in JJA dominate the figures. There is some limited GW activity visible in the tropics, for instance in DJF, but it is clearly minor compared to the polar regions. Gravity waves have been known to trigger PSC formation above mountains for quite some time, especially ice PSCs (see for instance https://doi.org/10.5194/acp-4-1149-2004). Trying to relate this GW activity with SICs will naturally lead to results dominated by PSCs. Again, it is a major problem to me that your paper does not address the confusion between stratospheric ice clouds in the Tropics (that can be generated by deep convection) and stratospheric ice clouds in the polar regions. PSCs are related to the polar vortex and are only partially represented in your dataset, both intentionally (as you filtered out some of them on purpose) and unintentionally (PSCs are optically thin and require specific detection and identification techniques). Given the object of study of your paper, I think it is necessary to make stronger efforts to exclude PSCs from the input dataset but also from the presentation of the results.

Thank you. To avoid the possible mixing of PSCs and SICs, we have excluded high latitude data in the new manuscript. Please see the answer to General Comment 2.

25. l. 385: Figure 10 is in my opinion way too busy to provide the basis for a reliable visual identification of correlations between stratospheric aerosols related to eruptions and SIC. It is too easy in my opinion to visually miss important features. Like with figures 3 and 6, the use of an Hovmoller plot is overkill and your interpretation would be much better served by zonal average plots.

We have revised this figure. For example, the high latitude data is removed from the figure, zonal mean information for the tropics and midlatitudes is added, the color palette is adjusted to proper ranges, and only strong volcanic eruptions are listed in the new plot. Please see Fig. 12. The Hovmöller plots are still used in the new manuscript because the latitude and color-palette optimized figures present more detailed information.

26. l. 413: Reverdy et al. 2012 do not use the CALIOP level 2 product that is used in the present article. Please clarify what misclassifications you imply, supported by more relevant references.

We have revised this subsection and added discussion on the 'misclassification'.

'The eruption of Puyehue-Cordón Caulle is known to have injected significant amounts of volcanic ash (Klüser et al., 2013; Hoffmann et al., 2014a). Moreover, Klüser et al. (2013) show that "the ash plume is transported very close to and potentially partly within or beneath ice clouds". In such a case the CALIOP "cloud fringe amelioration" algorithm might rather classify these detections as ice clouds instead of aerosol (Liu et al., 2019). Moreover, Liu et al. (2019) point out that the aerosol cloud classification for this volcanic plume was particularly challenging due to the dense and depolarizing aerosol.'

As volcanic aerosol is known to induce ice cloud formation and although the MIPAS data is more noisy and also shows some (not discussed) significant anomalies, which are not present in the CALIOP data, we consider the analysis of positive correlations between SICs and aerosol requires a more in-depths investigation to separate causal correlations from potential misclassifications in CALIPSO data.'

27. l. 428: "three regions..." how or why were these three regions selected? What makes them special a

priori? Why aren't the other regions worthy of their own correlation plots? Besides, it is unclear to me what the three full-page figures 13, 14 and 15 bring to our understanding of what drives the evolution of SIC. The figures-to-text ratio (3 pages of figures for less than one page of text) is off here. The text l.431-459 discusses when SIC occurrence is most correlated with a given metric and when it is correlated with another metric. Maybe only the metrics with an average correlation coefficient above a given threshold could be shown, or figures 13 to 15 could be moved to an appendix.

*We reorganized the structure and the regional analyses have been removed from the manuscript.*

28. l. 473 "solidly": Do you mean here that your analysis was good?

*We have revised this paragraph. 'In this study, a tropopause threshold of 250 m was applied to identify stratospheric ice clouds and stratospheric aerosols. As mentioned in Sect. 2.1, the vertical resolution of tropopause heights in ERA5 was improved by applying a cubic spline interpolation method (Hoffmann and Spang, 2022). When compared to radiosonde and GPS data, the height uncertainty for the LRT1 in ERA5 is less than 200 m (Tegtmeier et al., 2020a; Hoffmann and Spang, 2022).'*

29. l.473 "250m is a reasonable tropopause threshold..." again, I do not find this argument convincing.

*We have clarified the selection of tropopause threshold in the new manuscript. The vertical resolution of tropopause heights in ERA5 is improved after interpolating with cubic spline interpolation method (Hoffmann and Spang, 2022). And the height uncertainty for the LRT1 in ERA5 is less than 200 m compared to radiosonde and GPS data (Tegtmeier et al., 2020a; Hoffmann and Spang, 2022). Therefore, 250 m is used as the threshold to identify the stratospheric ice clouds. Please see answer to specific comment no. 3 and no. 28.*

30. Section 4.2: see my first major point.

*We have changed 'deep convection' to 'UTLS clouds'. 'In this study, temperature differences between AIRS brightness temperatures ($BT_{AIRS}$) and tropopause temperatures ($T_{TP}$) from ERA5 are employed to detect high altitude clouds in the tropics and at midlatitudes. A threshold of +7 K above $T_{TP}$ was chosen to identify possible high altitude clouds with tops in the upper troposphere and lower stratosphere, also referred to as UTLS clouds (Zou et al., 2021). In the tropics, most tropopause-reaching clouds with large optical thickness could be related to a deep convection origin (Gettelman et al., 2002; Tzella and Legras, 2011). At midlatitudes, high altitude clouds could also be related to frontal systems (warm front uplifting), mesoscale convective systems and mesoscale convective complexes, jet stream, mountain wave and contrails (Field and Wood, 2007; Trier and Sharman, 2016; Trier et al., 2020). UTLS clouds are considered here as a proxy for deep convection in the tropics and other high altitude ice cloud sources at midlatitudes.'*

31. l. 494-499: I think this has already been explained in lines 194-199.

*It is now removed from the manuscript.*

32. Figure 16: same remark as before on the limited usefulness of Hovmoller plots

*We revised the Hovmöller plots in Fig.12. The Hovmöller plots could present more detailed information than regional averaged linear plots after optimizing latitude range and color palette. Please see answer to specific comment no. 21.*

**References**

[revised manuscript text omitted]

---

## Author Comment (AC3)

**Reviewer #3**

**General Comments**

This manuscript presents a thorough evaluation of the global occurrence of stratospheric ice clouds (SICs) and their relationships with tropopause temperature, recent deep convection, gravity waves, and stratospheric aerosols. Potential relationships with modes of climate variability (QBO and ENSO) were also examined, but did not reveal strong association. The identification of SICs is based on the CALIPSO lidar observations, which are arguably the best available resource for global analysis, complemented by additional high-quality, satellite-based identification of the remaining parameters related to SICs. It was demonstrated well throughout that tropopause temperature and convection/gravity waves are strong controls for the occurrence of SICs, as expected. The analysis presented is thorough and the methods used are appropriate. It will be a meaningful contribution to the literature. I have one general comment related to an important element of the discussion that I believe is missing, but otherwise only a collection of suggested minor edits.

**Answer:** Thank you very much for reviewing our manuscript and for your comments and suggestions.

**General Comments**

an important limitation resulting from the use of CALIPSO data is bias introduced by the lack of sampling the diurnal cycle (critical for evaluating deep convection and, one would assume, much of the related SICs). I do not know how large the bias would be and am not aware of studies that allow one to quantify/estimate it well since it will be dependent not only on the frequency of sources, but also on the timescale of sublimation (which implies microphysics may be important, etc.). However, I do expect it is significant (especially over land masses). For example, overshooting convection in North America has a pronounced diurnal cycle with a frequency maximum that falls almost entirely between the CALIPSO sample times (e.g., see 10.1175/JAMC-D-15-0190.1, 10.1002/2017JD027718, 10.1029/2021JD034808). Other regions with frequent land-based overshooting storms have similar diurnal cycles. This suggests that much of the SICs that occur may not even be sampled, especially over land. Thus, the discussion throughout the manuscript requires an acknowledgement of this bias and how interpretations of the results might change.

**Answer:** Thank you for this helpful comment. We have added the discussion of the possible impact of the diurnal cycle of deep convection on the SIC detections from CALIPSO in Sect. 4.2.

'The sampling time of CALIOP may have an impact on the results presented here. While the diurnal cycle of high altitude reaching convection is well known (Hendon and Woodberry, 1993; Tian et al., 2006; Hohenegger and Stevens, 2013), little is known about the lifetime and diurnal cycle of SICs (Dauhut et al., 2020). At midlatitudes, over the central United States, the largest average fraction of overshoots was observed during the late afternoon to early evening local time (Cooney et al., 2018; Solomon et al., 2016), whereas CALIOP samples this area during the local minimum. In the tropics, the maximum precipitation from large mesoscale convective systems occurred in the local afternoon over land (Nesbitt and Zipser, 2003), but CALIPSO passes by the tropics after midnight (around 01:30 LT). Stratospheric clouds in the tropics have two peaks at 19:00–20:00 LT and the 00:00–01:00 LT from Cloud-Aerosol Transport System (CATS) lidar measurements. The expansion of convective clouds, the spread of winds, and the propagation of convective-generated gravity waves can all play a role in the high percentages of stratospheric clouds observed later (Dauhut et al., 2020). Since only measurements at 01:30 LT were used in this study, it is important to keep in mind the possible limitations associated with the diurnal cycles of deep convection and SICs.'

**Specific Comments:**

Line 17 - "and western" should be "and the western"

Fixed.

Line 37 – "hydrate stratosphere" should be "hydrate the stratosphere"

Fixed.

Line 40 – ", intensity" should be ", and the intensity"

Fixed.

Line 49 – "Six encounters…" the opening of this sentence is poorly phrased. Please revise.

We have revised the paragraph.

Line 72 – "ice clouds" should be "ice cloud"

Done.

Line 78 – "are" should be "is"

Fixed

Section 2.1 – I recommend defining the first and second lapse-rate tropopauses as LRT1 and LRT2, respectively. LRT is commonly used in the literature and helps to clearly communicate the definition used throughout. I would also recommend not bothering to define a cold-point tropopause acronym, since it is only used here.

Thank you. We have revised them throughout the entire manuscript.

Line 262 – recommend revising "motion of the Sun" to "location of peak insolation". The sun isn't moving…

We have revised this sentence.

Figure 2 – is this analysis relative to all observations or DT events only? Please add note to clarify.

We have revised in the manuscript. 'Occurrence frequencies of SICs associated with double tropopauses with respect to all profiles (a-d) and the fraction of SICs associated with double tropopauses to total SICs (e-h).'

Line 400 – "we" should be "were"

Fixed.

Figure 12 and related analysis – the overlap of several regions seems undesirable. It would be good to test sensitivity to having them be defined more exclusively.

Thank you. We have restructured the manuscript, and the regional analyses are removed from the manuscript.

Line 427 – "Spear-man" should be "Spearman"

Revised.

Line 448 – "Two" should be "The two" Line 480 – "in average" should be "on average" Line 486 – ",

lowering" should be ", and lowering"

Revised.

Line 527 – "In the MIPAS" should be "In MIPAS"

Fixed.

Line 588 – "tropauses" should be "tropopauses"

Fixed

Line 599 – "its" should be "their"

Fixed.

**References**

Cooney, J. W., Bowman, K. P., Homeyer, C. R., and Fenske, T. M.: Ten Year Analysis of Tropopause-Overshooting Convection Using GridRad Data, Journal of Geophysical Research: Atmospheres, 123, 329–343, https://doi.org/10.1002/2017JD027718, 2018.

Dauhut, T., Noel, V., and Dion, I.-A.: The diurnal cycle of the clouds extending above the tropical tropopause observed by spaceborne lidar, Atmospheric Chemistry and Physics, 20, 3921–3929, https://doi.org/10.5194/acp-20-3921-2020, 2020.

Hendon, H. H. and Woodberry, K.: The diurnal cycle of tropical convection, Journal of Geophysical Research: Atmospheres, 98, 16 623–16 637, https://doi.org/10.1029/93JD00525, 1993.

Hohenegger, C. and Stevens, B.: Controls on and impacts of the diurnal cycle of deep convective precipitation, Journal of Advances in Modeling Earth Systems, 5, 801–815, https://doi.org/10.1002/2012MS000216, 2013.

Nesbitt, S. W. and Zipser, E. J.: The Diurnal Cycle of Rainfall and Convective Intensity according to Three Years of TRMM Measurements, Journal of Climate, 16, 1456 – 1475, https://doi.org/10.1175/1520-0442(2003)016⟨1456:TDCORA⟩2.0.CO;2, 2003.

Solomon, D. L., Bowman, K. P., and Homeyer, C. R.: Tropopause-Penetrating Convection from Three-Dimensional Gridded NEXRAD Data, Journal of Applied Meteorology and Climatology, 55, 465 – 478, https://doi.org/10.1175/JAMC-D-15-0190.1, 2016.

Tian, B., Waliser, D. E., and Fetzer, E. J.: Modulation of the diurnal cycle of tropical deep convective clouds by the MJO, Geophysical Research Letters, 33, L20 704, https://doi.org/10.1029/2006GL027752, 2006.

---

## Author Response (AR2)

Dear editor and reviewer,

Thank you very much for handling and reviewing our manuscript and providing helpful comments. We considered all of the comments and revised the manuscript carefully. Please find our point-by-point replies below.

**Reply to the editor**

**Specific Comments:**

1. You should make clear from the outset that UTLS clouds as used in your work relate to the outcome of a specific data set. This would help to clarify the difference between UTLS clouds and SIC. IT should also be emphasized even stronger (already in the Abstract) that your definition of SIC excludes PSCs and what differentiates the two types of stratospheric clouds.

Thank you. We have added the explanation of UTLS clouds and the statement of PSC exclusion to the abstract.

'Ice clouds play an important role in regulating the water vapor and influencing the radiative budget in the atmosphere. This study investigates stratospheric ice clouds (SICs) in the latitude range between  $\pm 60^{\circ}$  based on the Cloud-Aerosol Lidar and Infrared Pathfinder Satellite Observations (CALIPSO). As polar stratospheric clouds include other particles, they are not discussed in this work. Tropopause temperature, double tropopauses, clouds in the upper troposphere and lower stratosphere (UTLS), gravity waves and stratospheric aerosols, were analyzed to investigate their relationships with the occurrence and variability of SICs in the tropics and at midlatitudes.

We found that SICs with cloud top heights of 250 m above the first lapse rate tropopause are mainly detected in the tropics. Monthly time series of SICs from 2007 to 2019 show that high occurrence frequencies of SICs follow the Intertropical Convergence Zone (ITCZ) over time in the tropics and that SICs vary inter-annually at different latitudes. Results show that SICs associated with double tropopauses, which are related to poleward isentropic transport, are mostly found at midlatitudes. More than 80 % of the SICs around  $30^{\circ}$ N/S are associated with double tropopauses.

Correlation coefficients of SICs and all the other above-mentioned processes confirm that the occurrence and variability of SICs are mainly associated with the tropopause temperature in the tropics and at midlatitudes. UTLS clouds which are retrieved from the Atmospheric Infrared Sounder (AIRS) and are used as a proxy for deep convection in the tropics and high altitude ice cloud sources at midlatitudes, have the highest correlations with SICs in the monsoon regions and the central United States. Gravity waves are mostly related to SICs at midlatitudes, especially over Patagonia and the Drake Passage. However, the second highest correlation coefficients show that the cold tropopause temperature, the occurrence of double tropopauses, high stratospheric aerosol loading, frequent UTLS clouds and gravity waves are highly correlated with the SICs locally. The long-term anomaly analyses show that inter-annual anomalies of SICs are more correlated with the tropopause temperature and stratospheric aerosols instead of the UTLS clouds and gravity waves.

The overlapping and similar correlation coefficients between SICs and all processes mentioned above indicate strong associations between those processes themselves. Due to their high inherent correlations, it is challenging to disentangle and evaluate their contributions to the occurrence of SICs on a global scale. However, the correlation coefficient analyses between SICs and all above-mentioned processes (tropopause temperature, double tropopauses, clouds in the upper troposphere and lower stratosphere (UTLS), gravity waves and stratospheric aerosols) in this study help us better understand the sources of SICs on a global scale.'

2. line 54: This statement is somewhat confusing. You mean that convective clouds inject ice directly rather than form ice clouds?

We have revised the sentence. 'Convective systems form ice clouds directly from anvil outflow, as well as indirectly from updrafts and wave-induced cooling'

3. lines 80-82: no need to introduce subsections in the introduction - line 87 and later: laps rates should be given in K/km

**Revised.** 'Section 3 presents the global SICs and relationship analyses between SICs and double tropopause, tropopause temperature, UTLS clouds, gravity waves, and stratospheric aerosols. Section 4 discusses the data uncertainties and relationship uncertainties between SICs and above-mentioned processes. Conclusions are presented in Section 5.'

4. line 89: criterion (singular)

Fixed.

5. line 118: Please be specific what's meant with high feature type quality.

The explanation is added to the manuscript. 'Samples marked with high feature type quality (the absolute value of the CAD score  $\geq$ 70) are used to ensure high reliability of data.'

6. line 125 and later: latitude range below 60 degree

Fixed.

7. Caption of Figure 1: please note in the caption what different line colours refer to and state the meaning of the hatched area

Thanks. We revised the caption. 'Occurrence frequencies of SICs on a  $5^{\circ} \times 10^{\circ}$  (latitude  $\times$  longitude) grid (a-d) and occurrence frequencies of ice cloud top heights (CTHs) in the altitude range from -4 to 4 km with respect to the first thermal tropopause (e-h) in DJF, MAM, JJA and SON from CALIPSO measurements. In e-h, the data are shown as zonal averages, globally (gray bars), for the tropics ( $20^{\circ}$  S -  $20^{\circ}$  N, orange lines) and midlatitudes ( $40^{\circ}$  - $60^{\circ}$  green lines for NH and purple lines for SH). The hatched areas are tropopause uncertainties of  $\pm 250$  m.'

8. line 227: should be 250 m or 0.25 km?

Revised, and the manuscript is unified using 250 m.

9. Figure 4 (and later): it might be nice to point out that the lines represent the data shown in Figure 1a-d?

We have added this explanation to all corresponding figures.

'Figure 4. Seasonal mean first tropopause temperature from ERA5 (color boxes) and occurrence frequencies of SICs from CALIPSO shown in Figure 1a-d (contour lines with an interval of 0.12).

Figure 5. Seasonal event frequency of UTLS clouds derived from AIRS during 2007-2019. Occurrence frequencies of SICs (data in Figure 1a-d) are shown in black contours with the interval of 0.12.

Figure 7. Mean brightness temperature variance at  $4.3 \,\mu$ m from AIRS measurements, which correlates with the amplitude of gravity waves. Black contours are occurrence frequencies of SICs shown in Figure 1a-d.'

10. Figure 7: Why are contours no longer in black as in the other plots?

We have updated all figures with black contours for SIC occurrence frequencies.

11. lines 301-310 and Figure 8 could be omitted. Alternatively, a correlation plot might be the better way to go here.

We have removed Figure 8 from the manuscript. The correlations between SICs and SAs are presented together with other processes in Section 3.7 for an overall analysis of all processes.

12. lines 323-326: should be moved to the caption of Figure 10

Revised.

**Reply to reviewer #1**

**General Comments**

I acknowledge the significant efforts made by the authors to improve their paper. Figure 11 is a nice addition. The article has been almost entirely rewritten – the track changes shows red or blue everywhere except in the bibliography. I am not currently able to re-review the entire article and check all the additions and changes, but based on the reply to reviewers and the parts of the revision I've read I would say that weak points remain in the manuscript and in some parts the writing appears rushed. The abstract especially needs work. I have comments below that I think still need to be addressed.

Note that I have identified problematic writing and thinking in the parts of the revision that were relevant to the original comments I raised, but there is no reason to believe that the rest of the revision is problem-free.

**Answer:** We appreciate your comments and suggestions for improving the manuscript, and we have revised the manuscript based on all the specific comments. Please see our point-by-point replies below.

**Specific Comments:**

1. abstract, line 12: "correlation coefficient ... indicate" – perhaps "confirm" would be better here. The fact that high clouds are driven by the tropopause temperature is not a new result.

Revised. 'Correlation coefficients of SICs and all the other above-mentioned processes confirm that the occurrence and variability of SICs are mainly associated with the troppoause temperature in the tropics and at midlatitudes.'

2. abstract, lines 12-15: You first say that the occurrence and variability of SIC are mainly associated with tropopause temperature in the Tropics. Then you say the Tropopause temperatures are \*mostly\* related to SIC in the midlatitudes – this implies that tropopause temperatures are \*less\* related to SIC elsewhere, for instance in the tropics (and you just said that they are). Please clarify by just stating upfront that SIC are primarily driven by tropopause temperature in both Tropics and midlatitudes.

We have revised those sentences.

'Correlation coefficients of SICs and all the other above-mentioned processes confirm that the occurrence and variability of SICs are mainly associated with the tropopause temperature in the tropics and at midlatitudes.'

'Gravity waves are mostly related to SICs at midlatitudes, especially over Patagonia and the Drake Passage.'

3. abstract, lines 14-15 : "besides the highest correlation coefficients" I don't understand what this means.

We have rephrased the sentence. 'However, the second highest correlation coefficients show that the cold tropopause temperature, the occurrence of double tropopauses, high stratospheric aerosol loading, frequent UTLS clouds and gravity waves are highly correlated with the SICs locally. The long-term anomaly analyses show that inter-annual anomalies of SICs are more correlated with the tropopause temperature and stratospheric aerosols instead of the UTLS clouds and gravity waves.'

4. abstract, lines 17-18: "The occurrence and variability of SICs demonstrate a strong dependence on various processes": Everything has a strong dependence on various processes. This sentence says nothing of value and can be omitted.

We have removed this sentence from manuscript.

5. abstract, line 19: "... and all processes" All processes means all processes, i.e. even those you have not considered in the paper. Please rephrase.

We have rephrased the sentence. 'The overlapping and similar correlation coefficients between SICs and all processes mentioned above indicate strong associations between those processes themselves.'

6. lines 100-106: This addition to the paper is very interesting and the justification very convincing. Thanks.

Thank you for this encouraging comment.

7. lines 123-125 in the revision, and answer to general comment 2: If you limit your analyses to the latitude range of  $\pm 60^{\circ}$ , there is not point in excluding clouds with CTHs above 12km at latitudes  $\frac{1}{60^{\circ}}$ . Please simplify your description.

We have revised this sentence to 'In this study, analyses are limited to the tropics and midlatitudes  $(\pm 60^\circ)$  to avoid interferences with the polar stratospheric clouds (PSCs).'

8. line 165: it is still unclear to me why stating that AIRS observations have the highest sensitivity at 30-40km requires referencing 3 Hoffmann papers. One would be enough.

We have retained only one reference in the revised manuscript.

9. line 214: "At midlatitudes, the frequencies of SICs at midlatitudes..." line 214: I don't understand "at least twice as low as in the Tropics".

We have revised this sentence. 'At midlatitudes, SICs are observed at least two times less frequently than in the tropics.'

10. line 201-204: I don't understand where the vertical distributions of ice clouds presented in figures 1eh come from. Are those results from CALIPSO? From AIRS? Are the plots showing the distribution of ice cloud top heights (as the vertical label suggests) or are those profiles of cloud fraction (as CALIPSO retrievals provide)? I am surprised that the midlatitude frequencies are mostly as large as the Tropics frequencies (while the maps show that SIC are much more frequent in the Tropics), are the distribution normalized? If they are you should say it. You need to better describe what is being shown here.

All the data presented in Figure 1 is from CALIPSO. Figure 1e-f shows the vertical distribution of ice cloud top heights. We have updated the figures and text in the manuscript. The similar occurrence frequencies of ice clouds in the tropics and midlatitudes in JJA can be affected by the location of ice cloud hotspots as the averaged values in the latitude range between  $20^{\circ}$  S -  $20^{\circ}$  N are presented in Figure 1g.

'Vertically, ice cloud top heights (CTHs) observed in CALIPSO are mostly found in the tropopause region ( $\pm$  500 m around the tropopause). Seasonally and regionally averaged occurrence frequencies of ice cloud top heights as a function of altitude are shown in Fig. 1 e-h for the tropics (20° S - 20° N) and midlatitudes (40° -60°N/S). Most ice cloud top heights are observed around the tropopause in the tropics and at midlatitudes. In the tropics, about 1 % of ice clouds have cloud tops 1 km above the tropopause in DJF, MAM, and SON. But very few ice clouds are found at midlatitudes with cloud tops 1 km above the tropopause. In JJA, relative low occurrence frequencies of ice clouds above the tropopause in the tropics can be affected by the location of ice cloud hotspots as we presented the averaged values in the latitude range between 20° S - 20° N.'

11. I'm still not sold on the value of some plots. Figure 2 still shows the evolution of SICs frequencies over 12 years but the discussion (lines 205-213) never mentions interannual variations, only the annual evolution, so what's the point? The discussion of Figure 12 (lines 365-389) is done through eyeball analysis which I find unreliable and unconvincing. This discussion would also be helped by referencing the figures when they are discussed (eg Fig 12a vs Fig 12c, etc). Figure 12d looks particularly useless, and is actually quite puzzling: the text references "high anomalies of UTLS clouds" that I have trouble seeing in the figure. Figure 12d suggests that UTLS clouds almost never happen, and that SIC necessarily originate from other sources. Please clarify.

We discussed the annual evolution of SIC occurrence frequencies in Figure 2 and the interannual variability of SIC anomalies in Figure 11 (Figure 12 in the previous manuscript). The 13 years of data shown in Figure 2 would present and confirm the seasonal features of SIC occurrence frequencies. We have revised the text and added sentences for this explanation. Figure 11 has been updated in the manuscript. We have deleted the UTLS cloud and gravity wave subplots because they do not correlate significantly with the SIC anomalies. The discussion in Figure 11 has been revised accordingly.

[revised manuscript text omitted]